EMBO
Molecular Medicine

# Lipid droplet-dependent fatty acid metabolism controls the immune suppressive phenotype of tumor-associated macrophages

Hao Wu[1,2,3,4], Yijie Han[5], Yasmina Rodriguez Sillke[2,3,6], Hongzhang Deng[7], Sophiya Siddiqui[2,3,4], Christoph Treese[2,3,8], Franziska Schmidt[2,3,4], Marie Friedrich[2,3,4], Jacqueline Keye[2,3,4], Jiajia Wan[1], Yue Qin[9] iD, Anja A Kühl[2,10], Zhihai Qin[1,*], Britta Siegmund[2,3,†] & Rainer Glauben[2,3,**,†] iD

## Abstract

Tumor-associated macrophages (TAMs) promote tumor growth and metastasis by suppressing tumor immune surveillance. Herein, we provide evidence that the immunosuppressive phenotype of TAMs is controlled by long-chain fatty acid metabolism, specifically unsaturated fatty acids, here exemplified by oleate. Consequently, en-route enriched lipid droplets were identified as essential organelles, which represent effective targets for chemical inhibitors to block *in vitro* polarization of TAMs and tumor growth *in vivo*. In line, analysis of human tumors revealed that myeloid cells infiltrating colon cancer but not gastric cancer tissue indeed accumulate lipid droplets. Mechanistically, our data indicate that oleate-induced polarization of myeloid cells depends on the mammalian target of the rapamycin pathway. Thus, our findings reveal an alternative therapeutic strategy by targeting the pro-tumoral myeloid cells on a metabolic level.

**Keywords** cancer immunotherapy; lipid droplets; lipid metabolism; tumor microenvironment; tumor-associated macrophage

**Subject Categories** Cancer; Immunology; Metabolism

## Introduction

Reprogramming of metabolic pathways guarantees the viability and proliferative capacity of cancer cells in a nutrient-poor environment (Pavlova & Thompson, 2016). These alterations include aerobic glycolysis (termed as Warburg effect; Warburg, 1956), increased glutamine uptake (Eagle, 1955), as well as amplified *de novo* fatty acid synthesis (Currie *et al*, 2013). Additionally, cancer cell-derived metabolites have been shown to contribute to the harsh tumor microenvironment (Anderson *et al*, 2017). In line, previous studies have already indicated decades ago that cancer cells but not stromal cells prefer to export fatty acids resulting in a fatty acid-enriched niche (Spector, 1967).

Tumor-associated macrophages (TAMs) represent an abundant population in the multi-cellular tumor microenvironment. In fact, the infiltration and differentiation of TAMs correlate positively with all stages of tumor progression (Noy & Pollard, 2014). Characterized as M2-like macrophages, TAMs exert multiple pro-tumoral properties. TAMs not only hamper anti-tumor immune responses via regulating T and NK cell activation and apoptosis, but also facilitate angiogenesis and metastasis (Lin *et al*, 2007; Gajewski *et al*, 2013; Yeo *et al*, 2014). Accordingly, macrophage-targeting therapies led to positive results in murine tumor models. Clodronate-liposome-mediated abrogation of macrophages, for example, resulted in limited tumor progression in a murine xenograft model (Zeisberger *et al*, 2006). Furthermore, the inhibition of macrophage recruitment, for instance, via CCR2 silencing, was followed by a significant inhibition of tumor development in both solid and hematologic tumor

1    The First Affiliated Hospital, Zhengzhou University, Zhengzhou, China
2    Corporate Member of Freie Universität Berlin, Charité - Universitätsmedizin Berlin, Humboldt-Universität zu Berlin, Berlin, Germany
3    Medical Department for Gastroenterology, Infectious Diseases and Rheumatology, Berlin Institute of Health, Berlin, Germany
4    Department of Biology, Chemistry, and Pharmacy, Freie Universität Berlin, Berlin, Germany
5    Institute of Biophysics, Chinese Academy of Sciences, University of Chinese Academy of Sciences, Beijing, China
6    Institute of Nutritional Science, University of Potsdam, Nuthetal, Germany
7    Department of Polymer Science and Engineering, Key Laboratory of Systems, Bioengineering (Ministry of Education), School of Chemical Engineering and Technology, Tianjin University, Tianjin, China
8    Berlin Institute of Health (BIH), Berlin, Germany
9    National Center for Nanoscience and Technology, Beijing, China
10   iPATH.Berlin – Core Unit of the Charité, Berlin Institute of Health, Berlin, Germany
     *Corresponding author. Tel: +86 371 6691 3632; E-mail: zhihai@ibp.ac.cn
     **Corresponding author. Tel: +49 30 4505 14343; E-mail: rainer.glauben@charite.de
     †These authors contributed equally to this work

models (Leuschner *et al*, 2011; Lesokhin *et al*, 2012). However, therapeutic strategies specifically targeting TAMs are still in early development stages (Cook & Hagemann, 2013).

Previous studies identified a distinct metabolic pathway between pro- and anti-inflammatory macrophages. Activated pro-inflammatory macrophages rely on glycolysis to meet the rapid energy consumption, while alternatively activated macrophages prefer to use fatty acid oxidation (Biswas & Mantovani, 2012; Galván-Peña & O'Neill, 2014). Considering the fatty acid-enriched tumor microenvironment and the anti-inflammatory phenotype of TAMs, we propose a model in which extracellular fatty acids polarize the infiltrating monocytes into M2-like pro-tumoral macrophages. Here, we found that fatty acids, especially unsaturated fatty acids, polarize bone marrow-derived myeloid cells into an M2-like phenotype with a robust suppressive capacity. Lipid droplets (LDs) play an essential role in this process by regulating the catabolism of free fatty acids (FFA) for mitochondrial respiration. Mammalian target of the rapamycin (mTOR) inhibition eliminates LD-derived mitochondrial respiration and therefore immune suppression, indicating a regulatory role of the mTOR signaling pathway in this process. Furthermore, both intra-tumoral injection of LD-associated inhibitors and specific disruption of LD formation in myeloid cells via liposome-mediated delivery system attenuated tumor growth in an *in vivo* model. Finally, analysis of colon cancer patients confirmed the correlation between the accumulation of LDs in TAMs and the clinical stage of tumor. Our results provide a novel mechanism as well as a therapeutic target on myeloid cell differentiation and therefore on tumor escape from immune surveillance.

# Results

## Oleate-induced mitochondrial respiration regulates the suppressive phenotype of myeloid cells

Previously, we identified a Gr1$^-$CD11b$^+$ subset within oleate-polarized myeloid cells with a potent T-cell suppressive capacity in the presence of granulocyte–macrophage colony-stimulating factor (GM-CSF; Wu *et al*, 2017). Here, we show that the Gr1$^-$CD11b$^+$ subset promotes tumor growth *in vivo* (Appendix Fig S1). To characterize this Gr1$^-$CD11b$^+$ population, fatty acid-treated myeloid cells were sorted and analyzed by microarray (Fig 1), which included further Gene Ontology (GO) analysis (Appendix Table S1). Interestingly, the expression level of differentially expressed genes in stearate-treated myeloid cells was close to the bovine serum albumin (BSA) control group, which differed significantly from the oleate-treated group, indicating a unique effect of oleate but not stearate on myeloid cell differentiation (Fig 1A). As expected, the core genes associated with *de novo* fatty acid synthesis and desaturation, for instance, *Fasn* as well as *Fads2*, *Fads3* and *Scd1* were, down-regulated (Fig 1B) when oleate was added as an external source. Remarkably, the LD formation-related genes, for instance, *Dgat1* and *Agpat9*, were up-regulated. As GM-CSF could functionally polarize dendritic cells (DC) *in vitro*, the DC signature was analyzed first: the classical DC signature led to 13 out of 24 identified overlapping transcripts while indeed all of them were down-regulated in the oleate-treated group compared to the BSA control group (Fig 1B; Miller *et al*, 2012). Essential transcription factors

including *Flt3* and *Btla* that regulate the maturation of DCs were also down-regulated. Hence, we concluded that oleate potently suppresses GM-CSF-induced DC polarization on a transcriptional level. Former work from Herber *et al* (2010) revealed that LD containing DCs failed to present antigen. Indeed, 14 MHCII complex-associated genes including *Ciita*, the master regulator for MHCII expression, were down-regulated in the oleate group, compared with either BSA or stearate treatment (Fig 1B; GO term Go: 42613; Go: 2504), which was confirmed by flow cytometry (Fig 1C). In contrast to DCs, five out of 14 mature macrophage signature genes were differentially expressed, of which three were up-regulated (Gautier *et al*, 2012). From these data, we concluded that, following oleate treatment, the Gr1$^-$CD11b$^+$ population exhibits an immature macrophage phenotype. Flow cytometry staining of F4/80 and CD11b further confirmed this macrophage subset (Fig 1C). In addition, the innate immune response-associated genes, including *Oas1a*, *Oas2*, *Oas3*, *Ifi202b*, *Irf7*, and *Tlr9*, were all down-regulated, indicating a deficiency in anti-tumor immune response (Fuertes *et al*, 2013). Furthermore, a cluster of TAM-associated genes was up-regulated including the surface marker *Mrc1*, M2 phenotyping markers *Arg1*, *Retnla*, *Chil3*, as well as the functional markers *Vegfa* and *Mmp9* (Fig 1B). An increased expression of the conventional TAM marker CD206$^+$ as well as a robust arginase activity in the oleate-treated group, detected by flow cytometry, confirmed the mRNA expression analysis (Fig 1C). Recently identified surface markers which have been associated with the inhibitory phenotype of myeloid cells, including CD38 and CD73 (Beavis *et al*, 2012; Karakasheva *et al*, 2015), were also elevated in the oleate group as determined by microarray and flow cytometry (Fig 1C). Thus, these data indicate that during differentiation, oleate alone is sufficient to induce TAMs on a phenotypical and functional level.

Prostaglandin E$_2$ (PGE$_2$) acts as an immune suppressive factor in malignancy niches (Torroella-Kouri *et al*, 2009). Our microarray data indicate that oleate treatment elevated the expression of *ptgs1* (COX1), the main producer of PGE$_2$ in eukaryotes. Thus, we were wondering whether COX1 contributes to the suppressive capacity of oleate-polarized myeloid cells. Myeloid cells were treated with celecoxib, an inhibitor of COX1, which effectively impaired the suppressive function of oleate-treated myeloid cells through diminishing nitric oxide (NO) production and arginase activity (Appendix Fig S2A and B). However, expression of CD38, CD206, and MHCII did not alter, indicating that PGE$_2$ synthesis is downstream or independent of the oleate-induced differentiation cascade (Appendix Fig S2C). Fatty acid metabolism is required for membrane synthesis as well as for energy consumption during the polarization of myeloid cells. Indeed, purified Gr1$^-$CD11b$^+$ cells from the oleate-treated group revealed a significant increase of the mitochondrial respiratory capacity under both quiescent and stressed conditions (Fig 2A), as indicated by basal oxygen consumption, spare respiratory capacity, maximal respiration, adenosine triphosphate (ATP) production, and proton leak. In contrast, the basal level of glycolysis, as defined by extracellular acidification rate (ECAR), did not alter (Fig 2B). To determine the contribution of mitochondrial respiration in oleate-induced polarization of myeloid cells, the chemical inhibitor

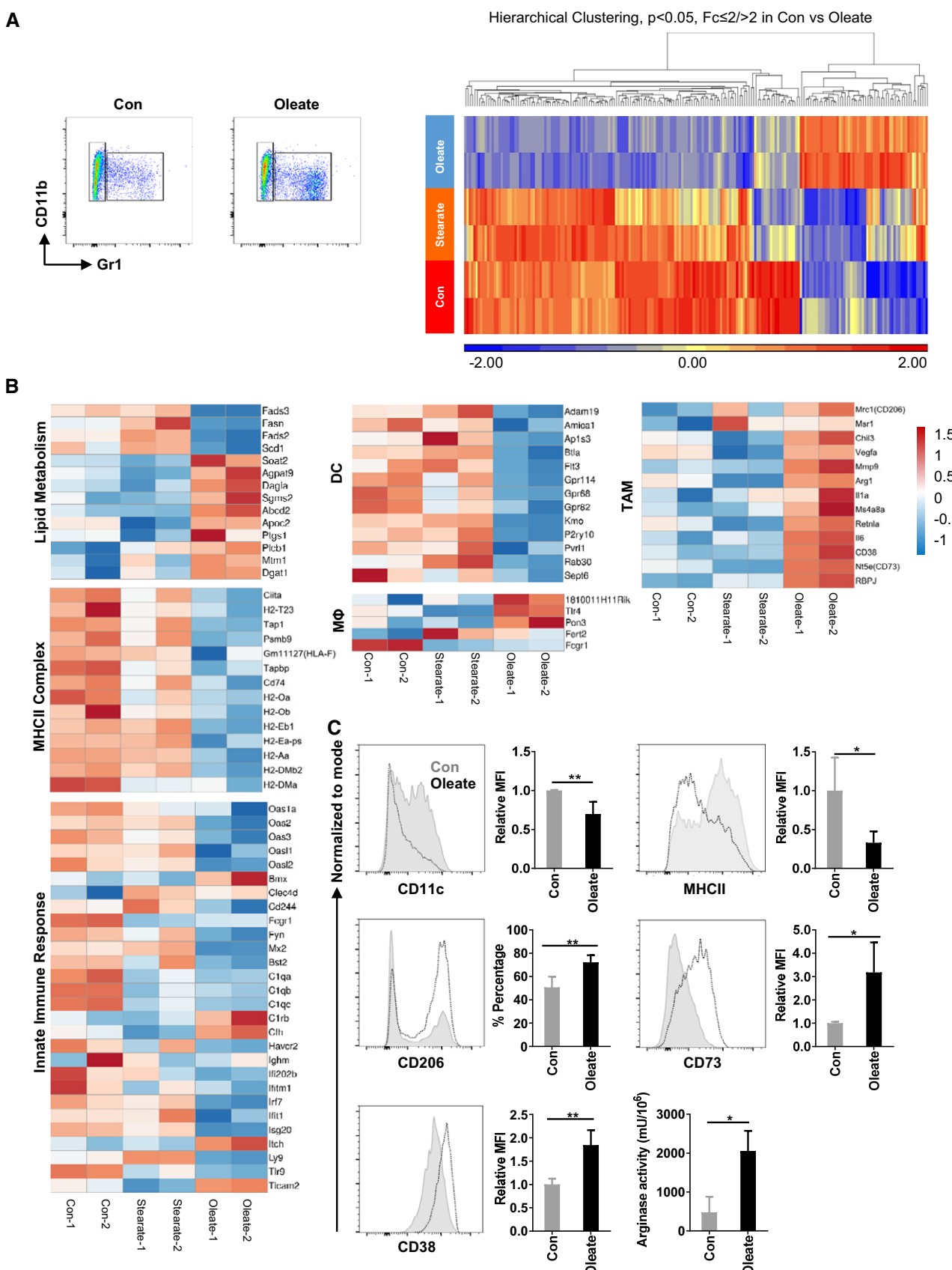

**Figure 1.**

◀

**Figure 1. Oleate polarizes bone marrow-derived myeloid cells into immune suppressive tumor-associated macrophages.**

A Bone marrow cells were polarized in the presence of 40 ng/ml GM-CSF and treated with 0.2 mM of the indicated compounds for 7 days. Gr1⁻CD11b⁺ cells were sorted and lysed for microarray. The hierarchical clustering was based on the different expression genes between BSA (Con) and oleate group.

B, C Signature genes involved in lipid metabolism, dendritic cell maturation, macrophage maturation, tumor-associated macrophages (TAMs) phenotype, MHCII complex, and innate immune response are listed (B) and validated (C) via flow cytometry or catalytic activity assay. Data are expressed as mean $\pm$ SD from two to four independent experiments. Unpaired Student's two-tailed $t$-tests were performed to compare the expression level of indicated proteins in control and oleate groups. $*P < 0.05$; $**P \leq 0.01$.

etomoxir was applied to block carnitine palmitoyltransferase 1 (CPT1), an enzyme associated with the outer mitochondrial membrane that transfers a long-chain acyl group from coenzyme A to carnitine, a process which is required to transport long-chain fatty acids into the mitochondrial matrix (Yao *et al*, 2018). Furthermore, it has been published that high concentrations of etomoxir (100 μM) directly impair mitochondrial respiration via decreasing the concentration of CoA in the cytosol or via inhibiting the mitochondrial respiratory complex I (Divakaruni *et al*, 2018; Yao *et al*, 2018). The treatment of etomoxir (40 μM) led to a reduction of mitochondrial respiration in both, the control and the oleate group (Fig 2C and D). Simultaneously, etomoxir exerted a comprehensive disruption of the oleate-induced effects, including the expression of CD206, CD38, and CD73 (Fig 2E and F), the inhibition of T-cell proliferation as well as NO production (Fig 2G–I), confirming the vital role of fatty acid oxidation in driving the maturation of CD206⁺ cells. David E. Sanin proved recently that PGE2 functions as a negative regulator of mitochondrial respiration in TAMs through modulating the malate–aspartate shuttle, thus explaining, why the reduction of PGE2 via COX1 inhibitor did not affect TAMs in our system (Sanin *et al*, 2018). Stearoyl-CoA desaturase-1 (SCD1) plays a crucial role in the endogenous production of unsaturated fatty acids. The SCD1 inhibitor CAY10566 served to estimate the contribution of endogenous unsaturated fatty acids on the polarization of CD206⁺ TAMs. Our data revealed that CAY10566 functionally impaired mitochondrial respiration and hampered the polarization of CD206⁺ myeloid cells in the control but not in the oleate group, indicating a compensatory effect of extracellular oleate (Appendix Fig S3). In summary, unsaturated fatty acid-derived mitochondrial respiration maintains the phenotype and function of TAMs *in vitro*.

### Lipid droplet-derived fatty acids facilitate mitochondrial respiration in myeloid cells

Our previous data demonstrated an intimate correlation between the enrichment of LDs and the suppressive phenotype of MSC-2 cell line as well as primary myeloid cells (Wu *et al*, 2017). Thus, we proposed that LDs act as a stable source of fatty acids to maintain the regulatory phenotype of macrophages. Diacylglycerol O-acyltransferase (DGAT) is responsible for the import of FFAs into LDs, while a cascade of lipases including adipose triglyceride lipase (ATGL), hormone-sensitive lipase (HSL), and monoacylglycerol lipase (MAGL) facilitate the depletion of LDs upon cell activation. Therefore, ATGL and HSL translocate to the LD membrane and cleave fatty acids from the stored triglycerides and therefore control the degradation of LDs. MAGL converts monoacylglycerols to the FFA and glycerol (Smirnova *et al*, 2006; Wang *et al*, 2009; Nomura *et al*, 2010; Fig 3A). We uncovered that inhibition of the catalytic activities of DGAT, ATGL, and MAGL attenuated oleate-induced mitochondrial respiration, especially ATP production (Fig 3B), as

well as the expression of CD206 and their immune suppressive capacity (Fig 3C–F). These data indicate that in fact the export of FFAs from LDs controls the polarization of suppressive myeloid cells. Thus, we identified LDs as a novel source of fatty acids contributing to the polarization of TAMs.

### mTORC signaling pathway-associated polarization of suppressive myeloid cells

Several signaling pathways orchestrate the lipid metabolism in eukaryotes. The mTOR signaling pathway has been proven as a master regulator of cellular metabolism (Wullschleger *et al*, 2006). Rapamycin is used widely to decipher the role of the mTOR complex, especially mTOR complex 1 (mTORC1), in various metabolic processes. Here, treatment with the mTOR inhibitor rapamycin functionally inhibited the maturation as well as suppressive function of the Gr1⁻CD11b⁺ population in both the control and the oleate-treated group, indicating a central role of mTOR signaling on suppressive myeloid cell differentiation and polarization (Fig 4A and B). Accordingly, rapamycin treatment decreased also mitochondrial respiration in both control and oleate-polarized myeloid cells (Fig 4C). Although originally established as a mTORC1 inhibitor, long-term treatment with rapamycin also affects the activation of mTORC2, due to the elimination of mTOR protein (Sarbassov *et al*, 2006). One method to distinguish between mTORC1 and mTORC2 activation is to detect the phosphorylation at specific sites of mTOR: serine 2448 for mTORC1 and serine 2481 for mTORC2, respectively (Copp *et al*, 2009). Phosphorylation analyses via flow cytometry and Western blot revealed a strong activation of mTORC1 in control but not in oleate-polarized myeloid cells (Fig 4D and E). In contrast, serine 2481 of mTORC2 was hyperphosphorylated upon oleate treatment (Fig 4E). Those data revealed a potent regulatory role of mTOR, especially mTORC2, in oleate-induced polarization of myeloid cells.

### Targeting lipid droplets in macrophages impairs tumor growth *in vivo*

*In vivo* data from MCA205 and CT26 inoculated tumor models indicate that tumor-infiltrating CD206⁺CD11b⁺ myeloid cells maintain the highest level of LDs compared to other immune cell populations (Fig 5A). To monitor the anti-tumor effect of DGAT inhibitors (iDGAT) *in vivo*, CT26 tumor-bearing mice were treated with vehicle or iDGAT starting at day 6 after tumor cell inoculation. As expected, iDGAT treatment effectively impeded tumor growth, while decreasing the proportion of CD206⁺ TAMs in CD11b⁺ cells in the tumor (Fig 5B and C). However, a direct effect of iDGAT on tumor cells could not be excluded. Studies reported a potent role of lipid metabolism in the survival and drug resistance of cancer cells (Boroughs & DeBerardinis, 2015; Iwamoto *et al*, 2018). Thus, to specifically

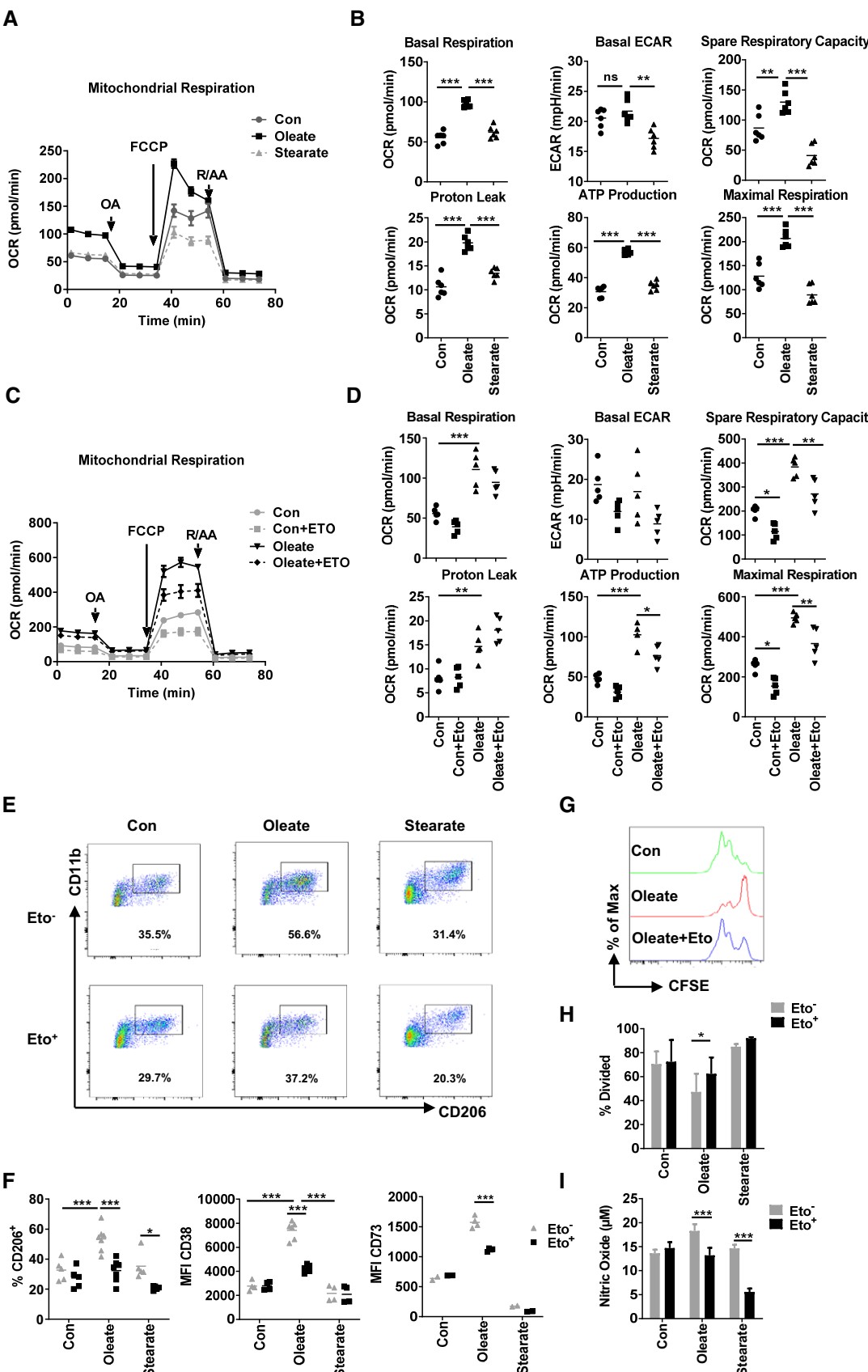

Figure 2.

**Figure 2.  Oleate-induced mitochondrial respiration regulates the suppressive phenotype of myeloid cells.**

A, B  Gr1⁻CD11b⁺ cells, polarized by the indicated treatment, were purified for mitochondrial respiration detection. The oxygen consumption rate (OCR) of these cells was monitored after the addition of oligomycin (OA; 1 μM), carbonyl cyanide-4-(trifluoromethoxy) phenylhydrazone (FCCP, 1 μM), and the electron transport inhibitor rotenone and antimycin A (R/AA; 0.5 μM) at indicated time points. The basal OCR, basal extracellular acidification rate (ECAR), spare respiratory capacity, proton leak, ATP production, and maximal respiration based on the OCR value were quantified.

C, D  Forty micromolar etomoxir was added starting from day 0 of bone marrow polarization, followed by mitochondrial respiration assay by using the same amount of polarized Gr1⁻CD11b⁺ myeloid cells on day 7. The mitochondrial respiration was monitored and analyzed via XFe96 Analyzer. The basal OCR, basal extracellular acidification rate (ECAR), spare respiratory capacity, proton leak, ATP production and maximal respiration based on the OCR value were quantified.

E, F  Exemplary plots of the proportion of CD206⁺ cells, the mean fluorescence intensity (MFI) of CD38 and CD73 on polarized myeloid cells was determined via flow cytometry.

G–I  T-cell proliferation assay was performed via co-culture with purified CD4⁺ T cells in the ratios of M (myeloid cells): T (T cells) = 1:30 (G, H). (I) Nitric oxide (NO) production from the co-culture supernatant was quantified by Griess reaction.

Data information: Data are expressed as mean ± SD from two to four independent experiments and analyzed by either one-way analysis of variance (ANOVA) (B) or two-way ANOVA with Tukey's *post hoc* test (D, F, H, I). *$P < 0.05$; **$P \leq 0.01$; ***$P \leq 0.001$.
Source data are available online for this figure.

disrupt LD formation in myeloid cells *in vivo*, peptide modified liposomes served as delivery vehicle for iDGAT treatment (Appendix Fig S4A) (Karathanasis *et al*, 2009). First *in vitro* experiments demonstrated that iDGAT-encapsulated liposomes functionally inhibit oleate-mediated LD formation in the suppressive myeloid cell line MSC-2, in murine splenic CD11b⁺ cells, and in human CD14⁺ monocytes (Appendix Fig S4B). In line, iDGAT-liposome treatment profoundly impaired MCA205 tumor growth (Fig 5D). Additionally, iDGAT-liposome treatment specifically reduced the amount of LDs in tumor-infiltrating myeloid cells but not in spleen or tumor-draining lymph nodes (Fig 5E). Consequently, the tumor-infiltrating CD8⁺ T cells were significantly reduced when the animals were treated with iDGAT compared to the untreated or vehicle-treated group (Fig 5F). Our data suggest that iDGAT treatment affects myeloid cells to hamper the infiltration of CD8 T cells into the tumor SCD1 inhibitor CAY10566 did not affect the tumor growth, which may be due to the compensation of extracellular fatty acids. Etomoxir, the inhibitor of CPT1, in fact hampered mitochondrial respiration *in vitro*, but only a tendency of delayed tumor growth could be detected, as its effective dose is above the loading capacity of liposome. In summary, we found a novel LD-related mitochondrial respiration-immune suppression (LMS) axis in mice, which promotes tumor escape from immune surveillance.

**Lipid droplets control the suppressive function of human PBMCs and correlate to human colorectal cancer progression**

Next, we wanted to know whether this novel LMS axis is relevant to human monocyte polarization. Since in bone marrow, not only Gr1⁻ stem cells, but also Gr1⁺ monocytes represent a source of CD206⁺ suppressive cells (Appendix Fig S5), we here used human monocytes to elucidate the effect of oleate. Purified CD14⁺ monocytes from healthy donors were treated with saturated or unsaturated fatty acids in the presence of GM-CSF. Flow cytometry data showed that oleate-treated monocytes elevated the expression of CD206, CD204, and CD38 on protein level (Fig 6A). Consistently, oleate treatment induced strong LD formation in human CD14⁺ monocytes as well as enhanced mitochondrial respiration and, most important, suppressive function (Fig 6B–D). Remarkably, iDGAT treatment completely reversed the oleate-induced suppression in these human myeloid cells (Fig 6D). Next, the correlation between LD-bearing macrophages and tumor progression in colorectal cancer (CRC) and gastroesophageal (G/E) cancer patients was evaluated via immunohistochemistry. Here, adipose differentiation-related protein (ADRP), a LD binding protein, served to identify and quantify LDs. Our data revealed a notable increase of LDs in CD68⁺CD206⁺ tumor-infiltrating myeloid cells in CRC patients when compared with adjacent non-tumor tissue (Control, Fig 6E). In contrast, G/E patients exhibited only scattered and weak ADRP signals (Appendix Fig S6). Our results are in agreement with data from the human protein atlas, which revealed high ADRP levels in CRC but not G/E cancer (Uhlen *et al*, 2017). A previous study by Funada *et al* (2003) reported that infiltration of macrophages at the invasive margin correlates with an increased overall survival rate. However, data from Bailey *et al* (2007) demonstrated that the sole density of macrophage infiltration within the tumor did not suffice as a parameter to predict the prognosis of cancer. Here, we found that the number of infiltrating CD206⁺CD68⁺ macrophages did not alter between benign tissue and tumor tissue in CRC patients. However, the formation of LDs in TAMs was specifically found in tumor tissue, indicating that quality but not quantity of TAMs could serve as parameter to predict the prognosis of cancer.

**Figure 3.  Lipid droplet-derived fatty acids facilitate mitochondrial respiration in myeloid cells.**

A  The formation and utilization of lipid droplets in eukaryotes. Five micromolar combination of DGAT inhibitors (DGAT1 inhibitor A922500 and DGAT2 inhibitor PF-06424439), 40 μM ATGL inhibitor atglistatin (Atg), or 5 μM MAGL inhibitor MJN110 was added to the bone marrow polarization system in the presence of 40 ng/ml GM-CSF and indicated compounds for 7 days.

B  The oxygen consumption rate (OCR) of 1 × 10⁵ purified Gr1⁻CD11b⁺ cells was monitored as described in Fig 2. The basal OCR, basal ECAR (extracellular acidification rate), spare respiratory capacity, proton leak, ATP production, and maximal respiration based on the OCR value were quantified.

C  The polarization state was evaluated via the expression of CD206.

D  T-cell proliferation assay was performed employing co-culture with purified CD4⁺ T cell in the variant ratios.

E  The lipid droplet accumulation was determined by BODIPY staining. The percentage of divided cells and proliferation index was calculated.

F  Nitric oxide (NO) concentration in the co-culture supernatant was quantified by Griess reaction.

Data information: Shown is the mean ± SD from two to four independent experiments and analyzed by two-way ANOVA with Tukey's *post hoc* test. *$P < 0.05$; **$P \leq 0.01$; ***$P \leq 0.001$.

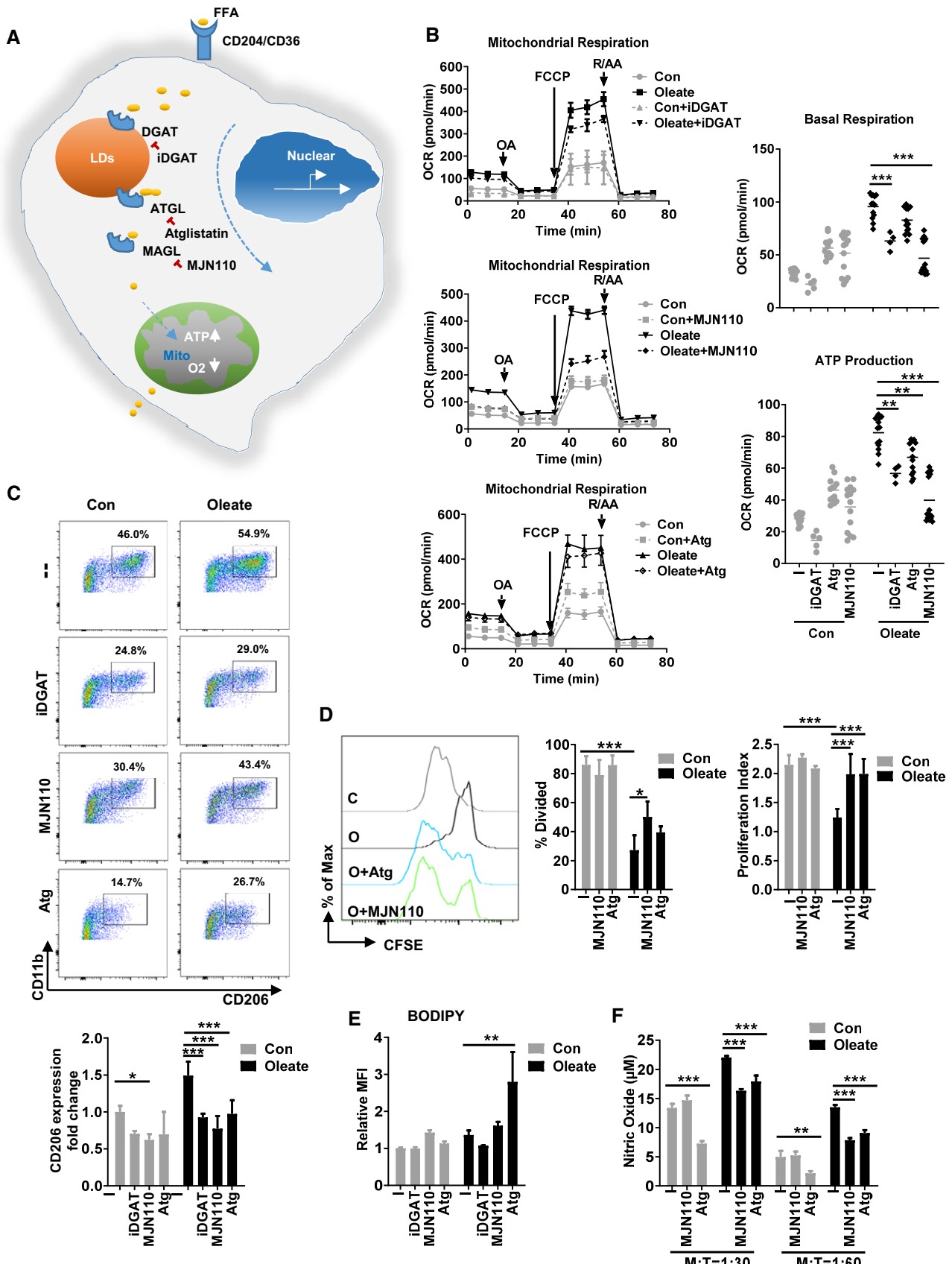

**Figure 3.**

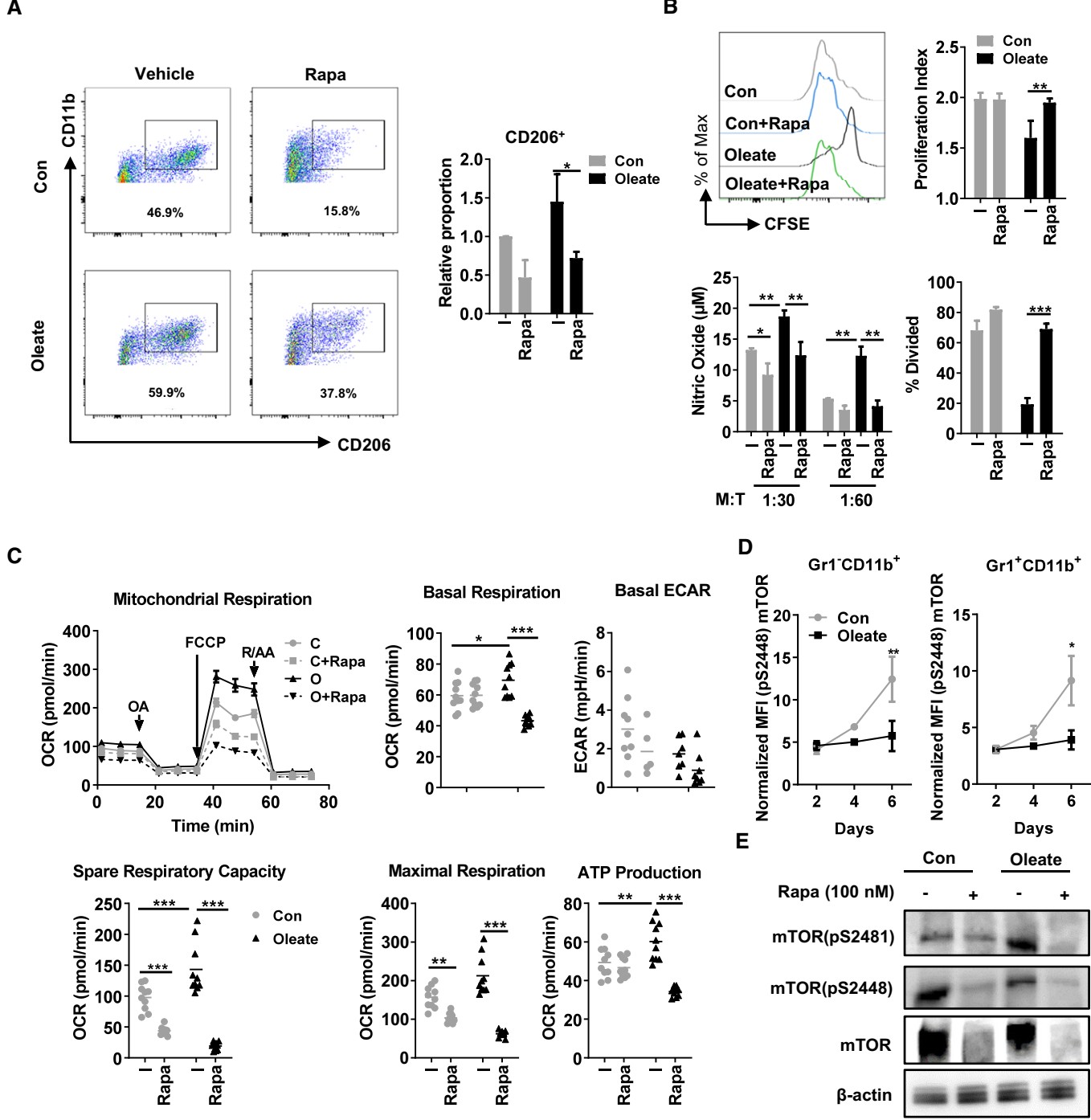

**Figure 4. Polarization of suppressive myeloid cells depends on mTOR signaling pathway.**

A    Bone marrow-derived myeloid cells were treated with BSA control (Con) or oleate in the presence or absence of 10 nM rapamycin (Rapa). The polarization of myeloid cells was detected via flow cytometry as indicated by CD11b and CD206 expression.

B    Purified Gr1⁻CD11b⁺ myeloid cells were co-cultured with CD4⁺ T cells for proliferation assay. The supernatant was collected for nitric oxide (NO) detection.

C    The oxygen consumption rate (OCR) of $1 \times 10^5$ differentiated myeloid cells were monitored after the addition of oligomycin (OA; 1 μM), the uncoupler carbonyl cyanide-4-(trifluoromethoxy) phenylhydrazone (FCCP, 1 μM), and the electron transport inhibitor rotenone and antimycin A (R/AA, 5 μM) at indicated time points. The basal OCR, basal extracellular acidification rate (ECAR), spare respiratory capacity, ATP production, and maximal respiration based on the OCR value were quantified.

D, E    Mean fluorescence intensity (MFI) of intracellular phosphor-mTOR (pS2448) in polarized myeloid cells as determined by (D) flow cytometry and (E) lysed for Western blot.

Data information: Shown is the mean ± SD from two to four independent experiments. Two-way ANOVA with Tukey's *post hoc* test was performed to compare the effect of rapamycin in different groups. *$P < 0.05$; **$P \leq 0.01$; ***$P \leq 0.001$.

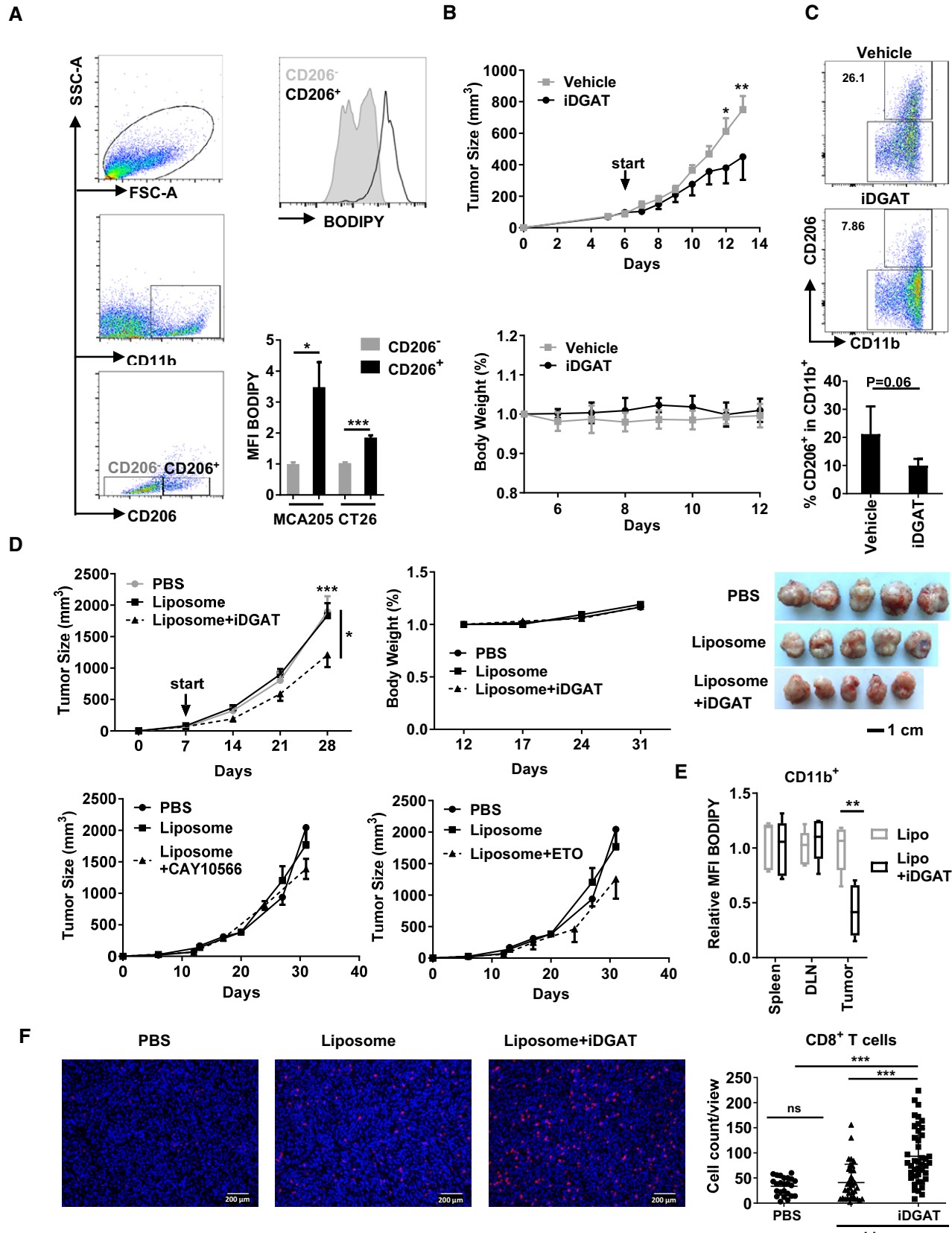

**Figure 5.**

◀

**Figure 5.   Disrupting lipid droplet-derived fatty acids in myeloid cells impairs tumor growth.**

A   Tumor-infiltrating immune cells were isolated from either CT26 or MCA205 tumor-bearing model and analyzed for the expression of BODIPY in CD206$^+$ (black) and CD206$^-$ (gray) cells.

B   Balb/c mice were inoculated with $5 \times 10^5$ CT26 tumor cells subcutaneously. Since day 6, vehicle or DGAT inhibitors (iDGAT) were administrated via intra-tumoral injection once daily.

C   At day 13, tumor-infiltrated immune cells were isolated and stained for CD206 and CD11b.

D   $5 \times 10^5$ MCA205 were injected subcutaneously into C57BL/6 mice. PBS, liposome control or iDGAT-liposome was injected into the peritoneum every other day starting at day 7. Tumor size and body weight were measured every 7 days and the mice were sacrificed on day 31 for analysis.

E   Immune cells from spleen, tumor-draining lymph nodes (TDLN), or tumor were isolated and stained for CD11b and with BODIPY for lipid droplet quantification in myeloid cells.

F   Tumor-infiltrating CD8$^+$ T cells (red) were stained together with nuclei (DAPI/blue) after sacrificing the tumor-bearing mice. White bars represent 200 μm.

Data information: Shown is the mean ± SEM from two to four independent experiments. Two-way ANOVA with Tukey's *post hoc* test was performed to compare the effect of iDGAT-liposome in tumor model. One-way ANOVA was performed to compare the effect of iDGAT-encapsulated liposome in human CD14$^+$ cells. *$P < 0.05$; **$P \leq 0.01$; ***$P \leq 0.001$.

## Discussion

The polarization of macrophages is a process known to be controlled by the reprogramming of cell metabolism. Previous studies indicated that the conventional anti-inflammatory cytokines used for polarization of M2-like macrophages are tightly connected to the lipid metabolism. For example, IL-10-controlled mitochondrial respiration has been shown to mediate the alternative activation of macrophages *in vitro* and *in vivo*. (Ip *et al*, 2017). However, it is still unknown whether the lipid metabolism itself plays an active role during the maturation of macrophages or whether the metabolic switch is just a consequence of the polarizing process. In the present study, we found that an unsaturated fatty acid-enriched environment affects the maturation of myeloid cells. We identified a CD206$^+$MHCII$^{low}$ suppressive myeloid cell population through oleate treatment alone. Those cells are distinct from classically characterized M1 and M2 macrophages, as they secrete high levels of TNFα, IL-6, and IL-1β, three typical M1 macrophage-derived cytokines (Wu *et al*, 2017). In fact, clinical data described a group of CD206$^+$MHCII$^{low}$ hypoxic TAMs in breast cancer patients with enhanced pro-angiogenic properties (Movahedi *et al*, 2010). Conventionally, TAMs maintain the signature of M2 macrophages and thus are termed as M2-like cells. In our study, those oleate-polarized CD206$^+$MHCII$^{low}$ myeloid cells were equally characterized by the expression of M2 markers such as Retnal, Arg1, Chile3l3, and CD206 on mRNA as well as on protein level (Fig 1). Furthermore, IL-6, MMP9, and VEGFα have been described as TAM markers (Chanmee *et al*, 2014), which also applied to our oleate-polarized CD206$^+$MHCII$^{low}$ regulatory myeloid cell population. Therefore, our results demonstrate that not only cytokine signaling modulates the phenotype of macrophages via metabolic reprogramming, but also enhancing the lipid metabolism itself is sufficient to induce CD206$^+$MHCII$^{low}$ immunosuppressive TAMs.

Whether or not the inhibition of mitochondrial respiration induces the impaired polarization of IL-4-induced M2 macrophage is under discussion. Divakaruni *et al* (2018) provided evidence that disruption of mitochondrial respiration via rotenone or antimycin did not impact the polarization of M2 macrophages. However, recent work from Huang *et al* (2014) proved a link between the M2 phenotype and oxygen consumption. In our study, all the inhibitors applied reduce ultimately the mitochondrial respiration as well as the expression of surface markers as CD206 and even more important the functional suppressive phenotype. Thus, based on our data we conclude that for our IL-4-independent model system, the described suppressive function is related to the mitochondrial respiration.

There are several resources for alternatively activated macrophages to maintain their supply of fatty acids. First, phagocytosis of extracellular fatty acids via surface receptors including CD36, Olr1 (LOX1), and CD204 is a steady way in a fatty acid-enriched environment, for instance, in a tumor microenvironment, explaining, that CD204 is suggested to be a marker of tumor-infiltrated macrophages (Ohtaki *et al*, 2010). In addition, recent studies described the expression of Olr1 in myeloid cells as indicator for their suppressive properties (Condamine *et al*, 2016). Second, lipases in the lysosome utilize also intracellular fatty acids. Stanley Huang *et al* (2014) have proven that the inhibition of lipolysis in the lysosome via tetrahydrolipstatin (Orlistat) impeded the alternative activation of macrophages, which then led to a decreased parasite elimination *in vivo*. The third resource is the *de novo* generation of fatty acids from redundant acetyl-CoA by fatty acid synthase. LDs are known to play a role in thermogenesis of adipocytes and muscle cells, yet little is known about the function of LDs in myeloid cells (Schreiber *et al*, 2017; Shan *et al*, 2017). Although the accumulation of LDs (also termed as lipid bodies) in macrophages has been investigated in infection models (D'avila *et al*, 2008), little is known about the contribution of LDs to the polarization of macrophages or their phenotype, respectively. Previous studies revealed that LD accumulation in DC resulted in a deficiency in antigen cross presentation (Herber *et al*, 2010). This phenomenon can be explained through the presence of oxidatively truncated lipids (ox-tr-LB) in LDs. The covalent binding of ox-tr-LB and chaperone heat-shock protein 70 blocks the translocation of pMHC to the cell surface, thus preventing DCs from presenting antigen to CD8 T cells (Veglia *et al*, 2017). In the present study, we found that both, the blockage of LDs formation via iDGAT and the degradation of LDs via ATGL and MAGL inhibitors, sufficiently inhibited the maturation, mitochondrial respiration, as well as suppressive function of myeloid cells. There is no unspecific effect known for any of the applied LD inhibitors, yet they all demonstrate a similar effect on myeloid cell polarization as well as inhibitory function, emphasizing that a pre-enrichment of fatty acids in LDs is a key process for the alternative activation of macrophages as well as their suppressive capacity. Consequently, liposome-encapsulated iDGAT effectively reduces tumor growth *in vivo* via both, intra-tumoral injection of iDGAT and systemic application of liposome-encapsulated iDGAT. Thus, we here define a novel therapeutic target as well as a potential drug

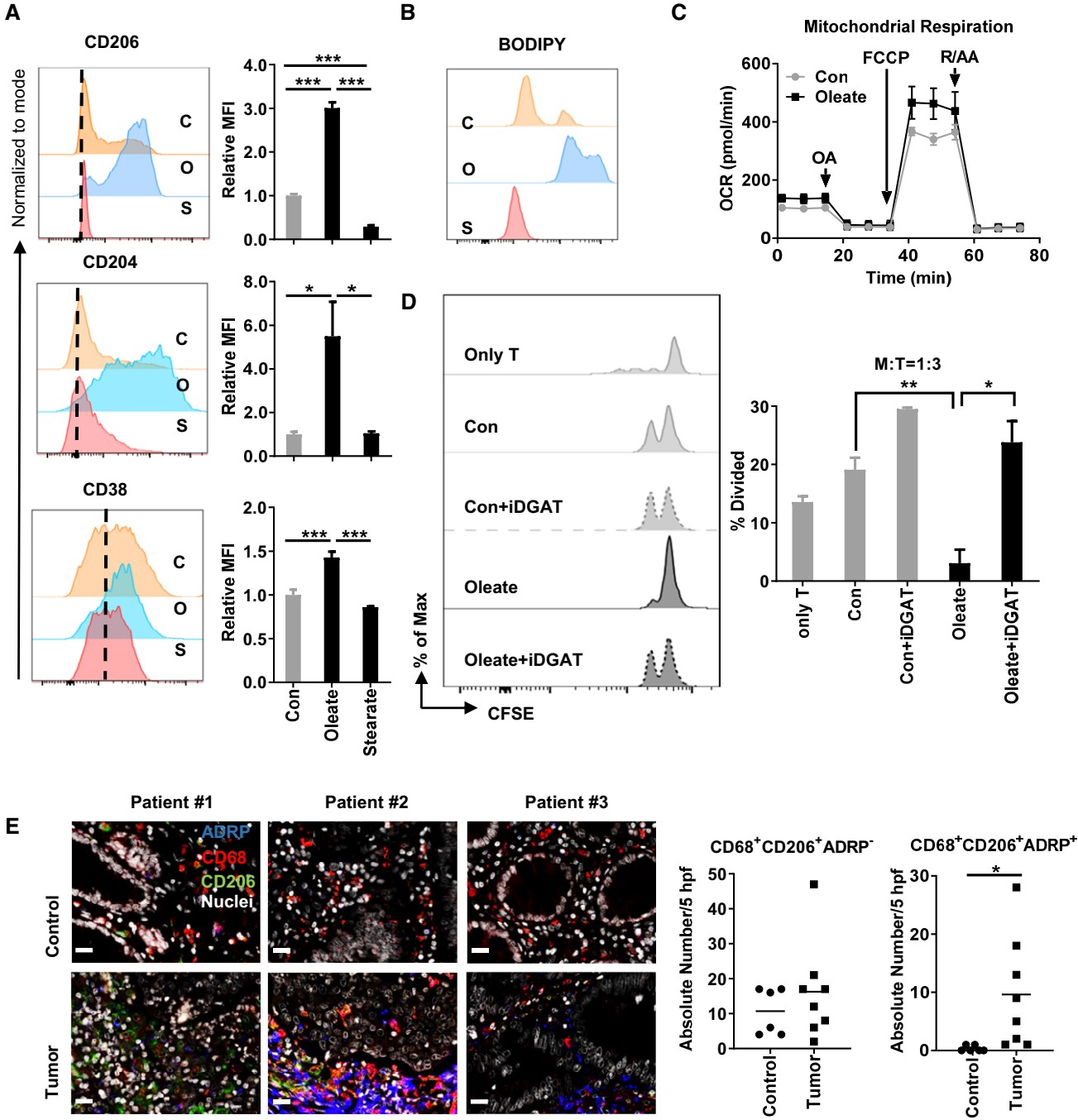

**Figure 6. Oleate induces polarization of human monocytes into CD206⁺ suppressive cells.**

A   $1 \times 10^6$ CD14⁺ myeloid cells from healthy donors were polarized by 1 ng/ml GM-CSF in the presence of BSA control (C), 0.2 mM oleate (O), or 0.2 mM stearate (S) over a 6-day period. The expression of CD206, CD204, and CD38 was analyzed via flow cytometry.

B   The level of lipid droplets was determined via BODIPY staining.

C   The oxygen consumption rate (OCR) of differentiated myeloid cells was monitored as described in Materials and Methods.

D   The purified CFSE-labeled autologous CD4⁺ T cells were co-cultured with polarized myeloid cells with indicated treatment for T-cell proliferation assay.

E   Tumor tissue (tumor) as well as corresponding non-tumorous adjacent tissue (control) was collected from colorectal cancer patients and prepared for histopathology. The expression of CD68 (red), CD206 (green), and ADRP (blue) indicates the lipid droplets in tumor-infiltrating myeloid cells. Nuclei (white) were stained using DAPI. The absolute number of positive cells was quantified in 5 high-power fields (hpf; scale bar = 20 μm).

Data information: Shown is the mean ± SD from two to four independent experiments and analyzed by either one-way ANOVA (A), Kruskal–Wallis test with Dunn's multiple comparisons test (D) or unpaired Student's two-tailed $t$-tests (E). *$P < 0.05$; **$P \leq 0.01$; ***$P \leq 0.001$.

(delivery) for tumor immune therapy by focusing on tumor-infiltrating myeloid cells.

The cellular metabolism is orchestrated by essential intracellular signaling cascades, for instance, by the mTOR signaling pathway. It has been described, that both, mTORC1 and mTORC2, respond to the polarization of M2 macrophages in parasite infection and tumor models (Covarrubias *et al*, 2016; Hallowell *et al*, 2017; Oh *et al*, 2017). It has been demonstrated, that the mTORC2-IRF4 axis controls the IL-4-induced M2 macrophage differentiation in a STAT6-independent way. Conversely, mTORC1 controls *de novo* lipid synthesis via the sterol responsive element binding protein transcription factors (Laplante & Sabatini, 2012), while the exact role of mTORC2 in lipid metabolism is not clear. Here, we found a strong activation of mTORC1 in the control group but not the oleate-treated group during the differentiation of bone marrow-derived myeloid cells. In contrast, the phosphorylation of mTORC2 was clearly triggered by oleate treatment. These data suggested an essential role of mTORC2 in the lipid metabolism-associated suppressive myeloid cell activation.

In summary, here we demonstrate that targeting the lipid metabolism of myeloid cells, especially LDs as critical cell structures, reverses the immune suppressive capacity of these cells *in vitro* and *in vivo*. Thus, a novel target, LD formation in myeloid cells, provides a novel anti-tumor strategy.

# Materials and Methods

### Cell lines

The murine colon carcinoma CT26 cells and the fibrosarcoma cell line MCA 205 were kindly provided by Prof. Zhihai Qin. The myeloid suppressor cell line MSC 2 cell line was a kind gift from Professor Ghiringhelli (Institut National de la Santé et de la Recherche Médicale [INSERM] U866, Dijon, France). All cell lines were tested for mycoplasma contamination on a monthly basis using Venor®GeM Advance Kit (Minerva Biolabs GmbH, Berlin, Germany) according to the manufacturer's protocols.

### Mouse tumor models

All animal protocols were approved by the regional animal study committee of Berlin (Germany) and the Institutional Animal Care and Use Committee of the Institute of Biophysics, Chinese Academy of Sciences. C57BL/6 and BALB/c mice (6–8 weeks of age, female) were purchased from either Janvier labs or the Weitong Lihua Company and were housed under standard conditions with free access to water and autoclaved standard chow. In co-injection tumor models, $5 \times 10^5$ CT26 cells were mixed with indicated amount of purified myeloid cells in Matrigel (BD Bioscience) and then injected subcutaneously. To test the anti-tumor effect of diacylglycerol acyltransferase (DGAT) inhibitor *in vivo*, intra-tumoral injection of DGAT1 inhibitor A922500 (3 mg/kg) and DGAT2 inhibitor PF-06424439 (10 mg/kg; both Sigma-Aldrich Chemie GmbH) were administered once daily at day 6 after $5 \times 10^5$ CT26 tumor cell inoculation. Tumor growth and body weight were monitored daily. To test the effect of LD formation inhibitors, $2.5 \times 10^5$ MCA205 cells were inoculated subcutaneously. Intraperitoneal injection of liposome-encapsulated inhibitors or liposome vehicle as control started at day 7. Tumor growth was monitored every 2–5 days, and tumor volume was calculated as (length × width × width/2).

### Fatty acid–albumin preparation

Fatty acid salts were dissolved in hot distilled water and added rapidly to proper cell culture medium with fatty acid-free BSA at a molar ratio of 8:1 (fatty acid salt:albumin). Fatty acid–albumin complex solutions were freshly prepared prior to each experiment.

### Affymetrix microarray analysis

Purified Gr1$^-$ myeloid cells were treated as indicated before (Wu *et al*, 2017) and total RNA was extracted by using the guanidinium isothiocyanate method (TRIzol reagent) followed by purification using the RNeasy Kit (Qiagen). RNA quality was assessed by using the Agilent Model 2100 Bioanalyzer (Agilent Technologies). Ten micrograms of total RNA was processed for analysis on the microarray by using the Affymetrix GeneChip one-cycle target labeling kit (Affymetrix) according to the manufacturer's protocols. The resultant biotinylated cDNA was fragmented and hybridized to the GeneChip Mouse Gene 2.0 ST Array. The arrays were washed, stained, and scanned using the Affymetrix Model 450 Fluidics Station and Affymetrix Model 3000 scanner according to the manufacturer's protocols. Expression values were generated by using Microarray Suite (MAS) v5.0 software (Affymetrix). Each sample and hybridization underwent a quality control evaluation, including cDNA amplification of > 4-fold, percentage of probe sets reliably detecting between 40 and 60% present call, and 3′–5′ ratio of GAPDH gene < 3.

### Evaluation and normalization of affymetrix GeneChip data

The hybridizations were normalized by using the robust multichip averaging method to obtain summary expression values for each probe set. Gene expression levels were analyzed on a logarithmic scale. For the assessment of the parental strains, regression, analysis of variance (ANOVA), *t*-tests, and the Mann–Whitney–Wilcoxon rank test were used to identify differentially expressed genes. Heat maps for gene expression data were generated by web tool Clustvis (Metsalu & Vilo, 2015).

### Inhibition of lipid droplet formation

The combination of iDGAT (DGAT 1 inhibitor: A922500; DGAT2 inhibitor: PF-06424439), ATGL inhibitor atglistatin (SML1075; Sigma-Aldrich Chemie GmbH) as well as MAGL inhibitor MJN110 (SML0872; Sigma-Aldrich) served to block the LD formation. Bone marrow-derived myeloid cells were cultured with the indicated FFA in the presence of variant inhibitors for 7 days.

### Liposome preparation

Liposomes were prepared by extruding a suspension of dissolved hydrated lipids through Whatman Nucleopore polycarbonate membranes in a Lipex Biomembranes Extruder (Northern Lipids, Vancouver, Canada). A lipid composition of 59.87% dipalmitoylphosphatidylcholine (DPPC), 40% cholesterol, and 0.12%

DSPE-PEG-peptide conjugate was used. The lipids were dissolved in ethanol at 60°C and then hydrated for 2 h with a 400 mM ammonium sulfate solution with ethanol ≤ 10% of final volume and 25 mM final lipid concentration. The suspension was passed four times with a 0.2-μm membrane and eight times with a 0.1-μm membrane through the extruder at 60°C and a pressure of approximately 100 psi. Liposomes were dialyzed (100 kDa MWCO) against phosphate-buffered saline (PBS) to establish an ammonium sulfate gradient. Liposome size was determined by a laser particle size analyzer (Zetasizer Nano) at a wavelength of 633 nm with a constant angle of 173°.

### Preparation of peptide conjugates

Here, we synthesized a cell type-specific binding of peptides (GGP peptide) for neutrophils and monocytes as conjugator for liposomes (Karathanasis *et al*, 2009). Briefly, 10.9 mg (5.6 μmol) of GGP peptide was dissolved in 500 μl of dimethyl sulfoxide (DMSO). DSPE-PEG (2000)-COOH (10 mg, 3.5 μmol), dicyclohexylcarbodiimide (DCC; 10.8 mg, 53.6 μmol), and 250 μl of pyridine were added to the reaction and incubated for 6 h at ambient temperature. Pyridine was removed by rotary evaporation. Conjugate micelles were formed by hydrating the DMSO solution with 5 ml of deionized water until DMSO reached 10% of the final volume. Unconjugated compounds were removed by dialysis (100 kDa MWCO) against 1 l, 50 mM NaCl (2×), and 1 l of deionized water (2×) followed by lyophilization. Final peptide content was determined using the DC protein assay (Bio-Rad, Hercules, CA, USA). The final conjugate was characterized by thin-layer chromatography on precoated plates (silica gel 60 F254) and MALDI-TOF mass spectroscopy (Applied Biosystems 4700 proteomics analyzer) to confirm molecular mass, with spectrum obtained in negative ion mode in a-cyano-4-hydroxycinnamic acid matrix.

### Loading of inhibitors into liposomes

Liposomes were loaded with inhibitors by an ammonium sulfate gradient. Ten milligram per milliliter A92250 (DGAT1 inhibitor) and PF-06424439 (DGAT 2 inhibitor) was encapsulated together into liposome at a ratio of 0.1 mg (A922500)/0.5 mg (PF-06424439) inhibitor per milligram of DPPC in the liposomes. Etomoxir and CAY10566 were loaded by the same procedure in the ratio of 0.1 mg (CAY10566)/0.5 mg (etomoxir) per milligram of DPPC. Liposome was cooled immediately on ice and dialyzed (100 kDa MWCO) twice against PBS to remove free inhibitors. The final inhibitor concentration after dialysis was determined after liposome lysis with 10% Triton X-100 by UV absorbance at 480 nm.

### Mitochondrial respiration analysis

Purified polarized $Gr1^-$ murine myeloid cells or human monocytes were seeded in XF96 cell culture microplates (Agilent) in at $1 \times 10^5$ cells/well in 180 μl pre-warmed assay medium. To allow media temperature and pH to reach equilibrium before the first rate measurement, cells were incubated at 37°C in 5% $CO_2$ for 2 h. Prior to each rate measurement, the XFe96 Analyzer (Seahorse Bioscience) gently mixed the assay media in each well for 3 min to allow the oxygen partial pressure to reach equilibrium. Following

mixing, oxygen consumption rate (OCR) and ECAR were measured simultaneously for 3–5 min to establish a baseline rate. The assay medium was then gently mixed again for 3–5 min between each rate measurement to restore normal oxygen tension and pH in the microenvironment surrounding the cells. After the baseline measurement, 20–27 μl of a testing agent prepared in assay medium was then injected into each well to reach the desired final working concentration. This was followed by mixing for 5–10 min to expedite compound exposure to cellular proteins, after which OCR and ECAR measurements were conducted. Generally, two to three baseline rates and two or more response rates (after compound addition) were measured, and the average of two baseline rates or test rates served for data analysis. Calculations were performed as follows: basal respiration = (last rate measurement before first injection) − (non-mitochondrial respiration), spare respiratory capacity = (maximal respiration) − (basal respiration), proton leak = (minimum rate measurement after oligomycin injection) − (non-mitochondrial respiration), ATP production = (last rate measurement before oligomycin injection) − (minimum rate measurement after oligomycin injection), maximal respiration = (maximum rate measurement after FCCP injection) − (non-mitochondrial respiration), non-mitochondrial respiration = minimum rate measurement after rotenone/antimycin A injection.

### Tumor-infiltrating immune cell

Briefly, tumor tissues were bathed in 70% isopropanol for 30 s and then transferred to a Petri dish. Tumors were minced into pieces < 3 mm in diameter and digested in 2 mg/ml collagenase type D at 37°C for 1 h. The digested tissue pieces were pressed through a 100-μm Falcon® nylon cell strainer (Corning, NY, USA). Purified cell suspensions were analyzed by flow cytometry.

### Bone marrow-derived myeloid cells

Bone marrow cells were isolated as previously described (Wu *et al*, 2017). Briefly, the cavities of femur and tibia bones of BALB/c mice were flushed with PBS. The single-cell suspensions were cultured in high glucose Dulbecco's modified Eagle's medium (4.5 g/l D-glucose) supplemented with 10% fetal calf serum, 100 U/ml penicillin, 100 μg/ml streptomycin, and 40 ng/ml GM-CSF (Peprotech 315-03). After 24 h of incubation, non-adherent macrophage progenitor cells were isolated and cultured in the presence of 0.2 mM BSA/sodium stearate or 0.2 mM BSA/sodium oleate plus 40 ng/ml GM-CSF for 7 days in six-well plates. Medium was renewed every other day.

### Murine T-cell proliferation assay

For the T-cell proliferation assay, syngeneic $CD4^+/CD8^+$ T cells were isolated from spleens and lymph nodes of healthy mice and purified via MACS (Miltenyi Biotec). The purity of $CD4^+$ and $CD8^+$ T-cell sorting was > 98%. T cells were stained with 0.5 μM carboxyfluorescein succinimidyl ester (CFSE) for 10 min at 37°C followed by thorough washing and resuspension in T-cell medium. Five microgram per milliliter concanavalin A or 96-well plates pre-coated with 5 μg/ml anti-CD3 (clone 2C11) plus 5 μg/ml soluble anti-CD28 (clone 37.51E1; both BD Pharmingen) served for T-cell

## The paper explained

### Problem

As cancer patients already benefit from strategies as CAR T-cell therapy or checkpoint inhibitors to (re)activate cytotoxic T cells, the overall response rate is still limited, especially in solid tumors. The tumor microenvironment has been considered a potential therapeutic target to enable an intrinsic anti-tumor response or to enhance the effect of immune cell-based anti-tumor treatments, respectively. As a main population in the tumor stroma and the main regulatory cell type in the microenvironment, tumor-associated macrophages (TAMs) are considered promising targets.

### Results

Our study reveals for the first time that the fatty acid-enriched environment itself is sufficient to induce the phenotype of TAMs, including the up-regulation of well-accepted markers like CD206, IL-6, VEGFα, MMP9, or Arg1. In addition, the very same LDs, identified in our *in vitro* polarization experiments upon sodium oleate treatment in myeloid cells, we found accumulated only in tumor-infiltrating TAMs but not in benign tissue of CRC patients.

Additionally, we found that the phenotype of oleate-induced TAMs, although termed "M2-like", is controlled by fatty acid oxidation, in a LD-dependent manner. Furthermore, we also provide evidence that this metabolism-driven TAM polarization is clearly mTORC2 associated.

Last, we identified DGAT, an enzyme responsible for the formation of lipid droplets in myeloid cells, as a potential anti-tumor immune therapy target. The oleate-induced polarization into immunosuppressive TAMs was prevented by cell-specific inhibition of DGAT *in vitro* in murine and human cell culture systems as well as *in vivo* in a murine tumor model.

### Impact

We here define the mechanism how unsaturated fatty acids polarize myeloid cells to the classical suppressive TAMs, delivering a new variety of potential pharmaceutical targets. Lipid droplets, an important source of fatty acids for mitochondrial respiration, are present specifically in TAMs of colorectal cancer but not in benign tissue and therefore appear as novel diagnostic parameter as well as potential therapeutic targets in tumor patients.

stimulation as indicated. Stimulated CFSE-labeled T cells were cultured in the presence or absence of beads-purified Gr1$^-$ bone marrow-derived myeloid cells in various ratios as indicated. After 72 h, the cultured cells were washed and gated on lymphocytes. The CFSE dilution due to cell division was analyzed by a FACSCanto II device (BD Biosciences) using FlowJo software (Tree Star, Inc).

### Nitric oxide measurements

The supernatant from T-cell proliferation assay was collected for NO measurements. NO production was defined as nitrite concentration using the Griess assay following the manufacturer's instructions (Promega).

### Arginase activity detection

Arginase activity was determined by measuring the amount of urea generated from the hydrolysis of L-arginine, as described previously (Zhao *et al*, 2012). Briefly, isolated cells were lysed in the presence of 100 μl lysis solution (0.1% Triton X-100, 10 mM MnCl$_2$, 25 mM Tris–HCl). The lysate of Gr1$^+$CD11b$^+$ cells was incubated with L-arginine, and the hydrolysis was stopped by adding 1 M sulfuric acid. Subsequently, α-isonitrosopropiophenone was added followed by boiling for 30 min. The concentration of urea was determined at 540 nm absorbance by the Infinite® F50/Robotic ELISA plate reader (TECAN, Männedorf, Switzerland). One unit enzyme activity was defined as the amount of enzyme that catalyzes the formation of 1 μmol of urea per min.

### Human monocyte polarization and T-cell proliferation assay

Peripheral blood samples were obtained from healthy volunteers, from which informed consent was obtained according to the local ethics committee of the Charité—Universitätsmedizin Berlin. All experiments conformed to the principles set out in the WMA Declaration of Helsinki and the Department of Health and Human Services Belmont Report. PBMCs were freshly isolated by density gradient centrifugation (Bicol; Biochrom). CD14$^+$ cells were magnetically labeled with CD14 MicroBeads (Miltenyi Biotec) and enriched with LS columns (Miltenyi Biotec). Isolated CD14$^+$ monocytes were taken into culture with RPMI-1640 (GIBCO, Life Technologies) supplemented with 10% fetal bovine serum (Invitrogen), penicillin/streptomycin (100 U/ml/100 μg/ml, Sigma-Aldrich Chemie GmbH), and with fatty acid-free BSA at a molar ratio of 8:1 (fatty acid salt: albumin) or additionally supplemented with 0.2 mM sodium oleate over 6 days. One nanogram milliliter GM-CSF was added. After 6 days, the CD14$^+$ monocytes were co-cultured with $3 \times 10^5$ purified autologous CD4$^+$ T cells for the T-cell proliferation analysis. Therefore, PBMCs were isolated from the same donor and CD4$^+$ cells were magnetically labeled with CD4 MicroBeads (Miltenyi Biotec) and enriched with LS columns (Miltenyi Biotec). The purity of CD14 and CD4 sorting was > 98%. T cells were stained with 0.5 μM CFSE for 5 min at 37°C followed by washing and resuspension in T-cell medium. CFSE-labeled T cells were cultured in a 96-well plate pre-coated with 1 μg/ml anti-CD3 (OKT3; BioLegend) and 1 μg/ml anti-CD28 (CD28.2; Invitrogen) together with the CD14$^+$ monocytes for 72 h in the ratio of 3:1. The flow cytometric analysis of the CFSE dilution due to cell division was performed by a FACSCanto II device (BD Biosciences) using FlowJo software (Tree Star, Inc).

### Flow cytometry

Flow cytometric analyses were performed using standard procedures. In brief, $2 \times 10^5$–$10^8$ cells were harvested and resuspended in 50 μl PBS/0.5% BSA buffer in the presence of the indicated antibodies for 10 min on ice. For boron-dipyrromethene (BODIPY) staining, cells were labeled by 0.2 μg/ml BODIPY®493/503 (Life Technologies) and incubated in 37°C for 15 min and washed with PBS. Data acquisition was performed using a FACSCanto II device (BD Biosciences) using FlowJo software (Tree Star).

### Western blot analysis

For detection of mTOR, phospho-mTOR (Ser2448) and phospho-mTOR (Ser2481; both Cell Signaling Technology), polarized myeloid cells were lysed in radioimmunoprecipitation assay buffer (RIPA buffer, Life Technologies, Carlsbad, CA, USA).

Western blot was performed via standard procedure. Densitometric analysis was performed with the Fuji MultiGauge software (Fujifilm).

### Histopathology

All human samples from CRC patients were kindly provided by the Central Biomaterial Bank Charité. Paraffin blocks of archived tissue samples were cut, dewaxed, and subjected to a heat-induced epitope retrieval step. Endogenous peroxidase was blocked by hydrogen peroxide prior to incubation with anti-CD68 (clone PGM-1, Dako) followed by the EnVision[+] System-HRP-Labelled Polymer Anti-Mouse (Dako). For visualization, OPAL-570 diluted in amplifier diluent (PerkinElmer) was used. Proteins were then inactivated and sections incubated with anti-CD206 (clone 5C11, LifeSpan Biosciences) followed by the EnVision[+] System-HRP-Labelled Polymer Anti-Mouse (Dako) and OPAL-520 diluted in amplifier diluent (PerkinElmer). After inactivation of proteins, sections were incubated with polyclonal anti-ADRP (#15294-1-AP, Proteintech) followed by the EnVision[+] System-HRP Labeled Polymer Anti-Rabbit (Dako) and OPAL-670 in amplifier diluent (PerkinElmer). Nuclei were stained using 4′,6-diamidine-2′-phenylindole dihydrochloride (DAPI; Sigma-Aldrich), and slides were cover-slipped in Fluoromount G (Southern Biotech). Multispectral images were acquired using a Vectra® 3 imaging system (Perkin Elmer). Positive cells were quantified in 5 high-power fields (field of vision in 400× original magnification). For CD8[+] T-cell detection, paraffin-embedded mouse tumor tissues were cut and dewaxed followed by heat-induced antigen retrieval. Sections were incubated with anti-CD8α (clone D4W2Z, CST, Germany), and PPD-540 digital image system (PANOVUE, Canada) was used following HRP-Labelled Polymer Anti-Rabbit (PANOVUE, Canada) conjugation for visualization. Nuclei were stained by using Fluoro-Gel II with DAPI (Electron Microscopy Sciences, PA, USA). All evaluations were performed in a blinded manner.

### Antibodies

All antibodies used in this study, including their working dilutions, are listed in Appendix Table S2.

### Statistical analysis

Statistical significance of differences between the experimental groups was determined by Student's *t*-test or factorial analysis of variance and the respective *post hoc* tests (Tukey's and Dunnett's multiple comparisons test) using GraphPad Prism software (GraphPad Software, La Jolla, CA). Exact *P*-values for all experiments can be found in a supplementary data sheet (Appendix Table S3).

## Data availability

Microarray data were submitted to the database of Gene Expression Omnibus (GEO) with the record number GSE118080 (https://www.ncbi.nlm.nih.gov/geo/query/acc.cgi?acc = GSE118080).

**Expanded View** for this article is available online.

## Acknowledgements

We are grateful to Inka Freise for her technical assistance. We thank the BCRT Flow Cytometry Lab (Berlin, Germany) and Désirée Kunkel for help with cell sorting. We also thank the BIH Core Facility Genomics (Berlin, Germany) and Ute Ungethüm for microarray and bioinformatics analysis. We thank also Gang Pei from the group of Stefan H. E. Kaufmann and Hongyu Liu from the Center 14 of Hematology and Oncology, Charité Campus Benjamin Franklin for their help with the pathway analysis or the mTOR phosphorylation studies, respectively. This publication was supported by grants from Helmholtz Alliance "Preclinical comprehensive cancer center" of the Helmholtz Society, the National Natural Science Foundation of China (81630068/31670881), the Deutsche Krebshilfe (70112011), the Deutsche Forschungsgemeinschaft (We5303-3-1), the National Natural Science Foundation of China (C0805/31600729), the China Postdoctoral Science Foundation, the China Scholarship Council, the "Fellowship for Young International Scientists"-program of the Chinese Academy of Science as well as the German-Israeli Foundation for Scientific Research and Development (1295).

## Author contributions

HW performed most of the experiments; HW, YH, and YQ performed animal experiments; HD synthesized and tested liposomes; YRS performed the human monocyte polarization and analysis; SS and JW performed revision experiments; CT provided G/E cancer samples; AAK performed the immunohistochemistry staining; HW, RG, and BS designed and analyzed the data; FS, JK, and MF revised the manuscript; HW, ZQ, BS, and RG wrote the manuscript.

## Conflict of interest

The authors declare that they have no conflict of interest.

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
