## [Review Process File · EMBO Molecular Medicine]

Lipid droplet dependent fatty acid metabolism controls the immune suppressive phenotype of tumor-associated macrophages

Hao Wu, Yijie Han, Yasmina Rodriguez Sillke, Hongzhang Deng, Sophiya Siddiqui, Christoph Treese, Franziska Schmidt, Marie Friedrich, Jacqueline Keye, Jiajia Wan, Yue Qin, Anja A. Kühl, Zhihai Qin, Britta Siegmund, Rainer Glauben

Review timeline:

Submission date:	2 April 2019
Editorial Decision:	26 April 2019
Revision received:	9 August 2019
Editorial Decision:	30 August 2019
Revision received:	9 September 2019
Accepted:	13 September 2019

Editor: Lise Roth

Transaction Report:

1st Editorial Decision

26 April 2019

Thank you for the submission of your manuscript to EMBO Molecular Medicine. We have now heard back from the 3 referees whom we asked to evaluate your manuscript.

As you will see from the reports below, while referees #2 and #3 are overall positive, referee #1 questions the validity of the experimental approach, and this point will need particular attention in a major revision of the present manuscript.

Addressing the other reviewers' concerns in full will also be necessary for further considering the manuscript in our journal. EMBO Molecular Medicine encourages a single round of revision only and therefore, acceptance or rejection of the manuscript will depend on the completeness of your responses included in the next, final version of the manuscript.

Please also contact us as soon as possible if similar work is published elsewhere. If other work is published, we may not be able to extend the revision period beyond three months.

I look forward to receiving your revised manuscript.

***** Reviewer's comments *****

Referee #1 (Comments on Novelty/Model System for Author):

This manuscript has a complete reliance on methods that have been shown to be non-specific and prone to experimental artifacts. In the absence of more rigorous experimentation, the conclusions are not justified by the experiments.

Referee #1 (Remarks for Author):

This manuscript describes the effect of unsaturated fatty acids on myeloid cells. They suggest that lipid droplet-derived fatty acids via mitochondrial fatty acid beta-oxidation mediate the polarization of myeloid cells. This manuscript and the methods herein are very similar to others work describing the requirement of fatty acid oxidation on macrophage and T-cell polarization. Unfortunately, those papers have been shown to be artifacts of the chemical inhibitors used in their studies. The exact same inhibitors have been used here. Therefore, the observations concerning the role of metabolism in instructing macrophage polarization here are almost certainly an artifact.

- Etomoxir Inhibits Macrophage Polarization by Disrupting CoA Homeostasis.

Divakaruni AS, Hsieh WY, Minarrieta L, Duong TN, Kim KKO, Desousa BR, Andreyev AY, Bowman CE, Caradonna K, Dranka BP, Ferrick DA, Liesa M, Stiles L, Rogers GW, Braas D, Ciaraldi TP, Wolfgang MJ, Sparwasser T, Berod L, Bensinger SJ, Murphy AN.

Cell Metab. 2018 Sep 4;28(3):490-503. PMID:30043752

- Loss of macrophage fatty acid oxidation does not potentiate systemic metabolic dysfunction.

Gonzalez-Hurtado E, Lee J, Choi J, Selen Alpergin ES, Collins SL, Horton MR, Wolfgang MJ.

Am J Physiol Endocrinol Metab. 2017 May 1;312(5):E381-E393. PMID:28223293

- Fatty acid oxidation in macrophage polarization.

Nomura M, Liu J, Rovira II, Gonzalez-Hurtado E, Lee J, Wolfgang MJ, Finkel T.

Nat Immunol. 2016 Mar;17(3):216-7. PMID: 26882249

Also, for example: Etomoxir is clearly not a specific inhibitor of Cpt1. See the below reference.

- Identifying off-target effects of etomoxir reveals that carnitine palmitoyltransferase I is essential for cancer cell proliferation independent of β -oxidation.

Yao CH, Liu GY, Wang R, Moon SH, Gross RW, Patti GJ.

PLoS Biol. 2018 Mar 29;16(3):e2003782. PMID: 29596410

Furthermore, the idea that lipid droplets are required for fatty acid oxidation has recently been shown to be incorrect as well.

- Cold-Induced Thermogenesis Depends on ATGL-Mediated Lipolysis in Cardiac Muscle, but Not Brown Adipose Tissue.

Schreiber R, Diwoky C, Schoiswohl G, Feiler U, Wongsiriroj N, Abdellatif M, Kolb D, Hoeks J, Kershaw EE, Sedej S, Schrauwen P, Haemmerle G, Zechner R.

Cell Metab. 2017 Nov 7;26(5):753-763.PMID:28988821

- Lipolysis in Brown Adipocytes Is Not Essential for Cold-Induced Thermogenesis in Mice.

Shin H, Ma Y, Chanturiya T, Cao Q, Wang Y, Kadegowda AKG, Jackson R, Rumore D, Xue B, Shi H, Gavrilova O, Yu L.

Cell Metab. 2017 Nov 7;26(5):764-777.PMID: 28988822

This manuscript has a complete reliance on methods that have been shown to be non-specific and prone to experimental artifacts. In the absence of more rigorous experimentation, the conclusions are not justified by the experiments.

Referee #2 (Comments on Novelty/Model System for Author):

This study investigated the role of lipid metabolism in regulating macrophage polarization through in vitro and in vivo experimental models. Most of the experiments are well-designed and conclusions were justified. These findings demonstrated the novel role of metabolic substrates, rather than canonical cytokines, in regulating the phenotypes and functions of tissue macrophages, and thus provide new insight into the field. They also showed that lipid droplets were found accumulated in CD68+CD206+ tumor infiltrating myeloid cells in CRC patients.

Overall, this is an interesting study with potential translational value.

Referee #2 (Remarks for Author):

In this study, Wu et al. investigated the role of lipid metabolism in regulating macrophage polarization through in vitro and in vivo experimental models. Their results showed that fatty acids, especially unsaturated fatty acids, polarized both mice and human myeloid cells into an M2-like phenotype. Unsaturated fatty acids induced mTOR phosphorylation, which activated lipid droplets catabolism and mitochondrial respiration, thereafter inducing an immunosuppressive M2-like phenotype in macrophages. They also showed that inhibitors antagonizing the above pathway could attenuate M2 polarization and inhibit tumor growth in vivo, and lipid droplets were found accumulated in CD68+CD206+ tumor infiltrating myeloid cells in CRC patients.

Overall, this is an interesting study with potential translational value. Most of the experiments are well-designed and conclusions were justified. These findings demonstrated the novel role of metabolic substrates, rather than canonical cytokines, in regulating the phenotypes and functions of tissue macrophages, and thus provide new insight into the field. The study could be further improved by addressing the following minor concerns:

1. Did you see the dose effect of oleate and stearate on the polarization of bone marrow-derived myeloid cells?
2. The results showed that oleate-exposed macrophages suppressed T cell proliferation, and inhibition of lipid droplets pathway in macrophages antagonized such inhibitory effects. Did they also affect the functional activity or markers on T cells?
3. In the discussion, the authors claimed that "analysis of colon cancer patients confirmed the correlation between the accumulation of LDs in TAMs and the clinical stage of tumor." However, these data are not found in the manuscript.
4. Same subtitles of the first and the second part of the results (page 5 and page 6)?
5. There are numerous typos, e.g., p14, line 22, "provides anovel anti-tumor strategies", and the manuscript should be carefully checked through.

Referee #3 (Comments on Novelty/Model System for Author):

The data presented by Wu et al provides relevant information in the field of immunometabolism and cancer by proposing a therapeutic strategy against pro-tumoral derived myeloid cells based on targeting the LD content which could be relevant and valuable in a clinical setting and probably translatable in the future

Referee #3 (Remarks for Author):

Review of the manuscript entitled " Lipid droplet dependent fatty acid metabolism controls the immune suppressive phenotype of tumor-associated macrophages "

The authors investigate the role of long fatty acid metabolism on the immunosuppressive phenotype of TAM. They perform in vitro and in vivo studies to demonstrate that the TAM polarization can be modulated by unsaturated fatty acids and the lipid droplet content. Moreover, the analysis of tumor infiltrating myeloid cells from human samples shows a correlation of the increased lipid droplets accumulation with the clinical stage of the tumor. They conclude targeting lipid droplets provides a therapeutic strategy against pro-tumoral myeloid cells.

The data presented by Wu et al provides relevant information in the field of immunometabolism and cancer by proposing a therapeutic strategy against pro-tumoral derived myeloid cells which could be relevant and valuable in a clinical setting and probably translatable in the future.

The article is well organized, clear and straightforward. The results are interesting and solid and the approaches are accurate and adequate to the answers the authors want to get. The in vivo studies together with the analysis of myeloid infiltrating cells from human colon cancer samples strengthens the conclusions.

Cell bioenergetics experiments elegantly demonstrate the crucial role of fatty acids mobilization from lipid droplets to sustain mitochondrial oxidative phosphorylation in TAM.

However, there is a minor comment that the authors should address before publication.

Minor comments:

.-pg 7: Replace: "In this context, carnitine palmitoyltransferase 1 (CPT1) controls the import of long chain free fatty acids into the mitochondria via converting coenzyme A into l-carnitine" by "CPT1a catalyzes the transfer of the acyl group of long-chain fatty acid-CoA conjugates onto carnitine, which is an essential step for the mitochondrial uptake of long-chain fatty acids for subsequent beta-oxidation in the mitochondrion".

1st Revision - authors' response

9 August 2019

***** Reviewer's comments *****

Referee #1 (Remarks for Author):

This manuscript describes the effect of unsaturated fatty acids on myeloid cells. They suggest that lipid droplet-derived fatty acids via mitochondrial fatty acid beta-oxidation mediate the polarization of myeloid cells. This manuscript and the methods herein are very similar to others work describing the requirement of fatty acid oxidation on macrophage and T-cell polarization.

We thank the reviewer for reading and reviewing our manuscript and for commenting on our data. Nevertheless, we would like to clarify, that we do not describe “the requirement of fatty acid oxidation on macrophage and T-cell polarization”. What we describe here, is an alternative pathway how macrophages polarize to suppressive cells in an IL-4-independent, but fatty acid and lipid droplet-dependent manner. More importantly, we proved in a mouse model that DGAT1 and 2 in myeloid cells represent potential targets in tumor therapy, which has not been published before.

Unfortunately, those papers have been shown to be artifacts of the chemical inhibitors used in their studies. The exact same inhibitors have been used here.

Our main message is not based on inhibitors of fatty acid oxidation, but on the effect of fatty acids on macrophages within the tumor microenvironment. With our data we provide for the first time evidence that DGAT inhibition could effectively block tumor growth by inhibiting the polarization the CD206+ suppressive myeloid cells. We, of course, are aware of potential side effect of chemical inhibitors. As proven by us and other groups, neither treatment with DGAT1 nor DGTA2 inhibitor alone could block the lipid droplets formation. Although genetic modification via shRNA or the CRISPR/Cas9 system might appear superior over chemical inhibitors, both systems also suffer from off target effects (especially when two genes have to be knocked down or knocked out). Additionally, both systems are still not ready for clinical intervention even treatment in animal models is still highly experimental. Homozygous DGAT2 knock out mice cannot survive after birth. Conditional knockout of both DGAT1 and DGAT2 in macrophages would be a more elaborate way to study these enzymes unfortunately they do not exist yet, but we are in the process of generating these mice for future studies. For these reasons, chemical inhibitors represent currently the best available tools to prove the anti-tumor effect of lipid-droplet inhibition in myeloid cells. A922500 and PF06424439 are specific DGAT1 and DGTA2 inhibitors. A922500 was used in the dose of 75 μM in Huh7-Lunet cells (1). PF06424439 was used in the dose of 10 μM in MEFs (2). Here, in our article we used 5 μM for both inhibitors. Despite the rather low concentration, it was sufficient to block the effect of oleate in myeloid cells. Although not impossible, we find it difficult to assume a side effect, mimicking the specific effect, which only appears if a combination of these two chemicals is applied. Furthermore, with regard to their chemical structure, neither A922500 nor PF06424439 is able to bind Coenzyme A. We emphasize the specificity now in the manuscript (page 14 line 9-11).

Therefore, the observations concerning the role of metabolism in instructing macrophage polarization here are almost certainly an artifact.

We agree again, that there are side effects of chemical inhibitors and acknowledge the mentioned side effect of etomoxir. However, it is extremely unlikely, that from the five chemical inhibitors, we used to decipher the respective pathway, every single one shows the same side effect resulting in the same macrophage phenotype. For instance, atglistatin, one of the inhibitors applied in our study, has been used in both thermogenesis publications cited by the reviewer (Schreiber et al. and Shin et al.) and was explicitly called “specific” in these publications.

We thank the reviewer for these references. We added and discuss these references now in the revised version of the manuscript (page 7 line 12 – page 8 line 10, page 13 line 4-9). As part of the mandatory point-to-point reply, I feel obliged to comment on these references in the revision letter at hand.

- Etomoxir Inhibits Macrophage Polarization by Disrupting CoA Homeostasis. Divakaruni AS, Hsieh WY, Minarrieta L, Duong TN, Kim KKO, Desousa BR, Andreyev AY, Bowman CE, Caradonna K, Dranka BP, Ferrick DA, Liesa M, Stiles L, Rogers GW, Braas D, Ciaraldi TP, Wolfgang MJ, Sparwasser T, Berod L, Bensinger SJ, Murphy AN. *Cell Metab.* 2018 Sep 4;28(3):490-503. PMID:30043752

This conclusive publication demonstrates clearly the unspecific effects of etomoxir on IL-4 and M-CSF polarized macrophages. Yet, we describe in our manuscript a GM-CSF-dependent, fatty acid (oleate)-induced and - even more important - IL-4-independent mechanism. Our cells were cultured over six days in the presence of high dose oleate and analyzed subsequently. In contrast, in the M2-protocol applied by Divakaruni et al (any many others), the macrophages were analyzed after a 24h stimulation with IL-4 for polarization. Therefore, metabolically and all the more immunologically speaking, these are different myeloid subtypes with different markers and different functions. Unfortunately, functional data are missing in the cited manuscript. Thus, a direct comparison is not possible.

- Loss of macrophage fatty acid oxidation does not potentiate systemic metabolic dysfunction. Gonzalez-Hurtado E, Lee J, Choi J, Selen Alpergin ES, Collins SL, Horton MR, Wolfgang MJ. *Am J Physiol Endocrinol Metab.* 2017 May 1;312(5):E381-E393. PMID:28223293

As with the publication above, we follow a different hypothesis: we do not polarize M-CSF-derived macrophages with IL-4. Main point of our story is the existence of an IL-4-independent way to polarize macrophages in the presence of GM-CSF and a FFA-rich environment as the tumor microenvironment. And as with the publication above, there are no functional data we could compare our data with. Additionally, most data are just mRNA data and the main marker we applied, and which is used widely for experiments or analyses regarding TAMs, CD206, is mostly missing. Interestingly, in the macrophage polarization in the gonadal fat tissue (reference Fig 5) where CD206 was measured on protein level, the frequency of CD206+ cells was reduced in the CPT2-KO-mice, even if n=5 seems not to be enough for a statistical significance. And when cells were treated with FFA (although, for macrophage polarization a rather ineffective oleate-to-palmitate-ratio of 2:1), a slight decrease of Cox2, Mcp1 and Arg1 could be overserved (n=6) in the CPT2-KO cells.

- Fatty acid oxidation in macrophage polarization. Nomura M, Liu J, Rovira II, Gonzalez-Hurtado E, Lee J, Wolfgang MJ, Finkel T. *Nat Immunol.* 2016 Mar;17(3):216-7. PMID: 26882249

As above, this publication tells a different story than our manuscript, but an interesting one nevertheless. We would like to point out that the etomoxir dose applied in this publication to prove the unspecific effect is even 5 times higher than ours.

Also, for example: etomoxir is clearly not a specific inhibitor of Cpt1. See the below reference.

- Identifying off-target effects of etomoxir reveals that carnitine palmitoyltransferase I is essential for cancer cell proliferation independent of \$\beta\$ -oxidation. Yao CH, Liu GY, Wang R, Moon SH, Gross RW, Patti GJ. *PLoS Biol.* 2018 Mar 29;16(3):e2003782. PMID: 29596410

The publication demonstrates (again), that etomoxir in high concentrations becomes unspecific and shows nicely why. Ultimately, they deal with a very different cell type (human breast epithelial cells), even a cell line and the concentration of etomoxir is again 5 times higher than what we used in our study. Overlap to our experiments at best: the etomoxir experiments in Fig. 2.

Please allow us to summarize the two main points, why we still are convinced, that the unspecific effect of etomoxir does not affect our conclusion:

1) IL-4 polarized M2 macrophage are different from oleate polarized suppressive myeloid cells.

IFN γ +LPS or IL-4 are classical methods to polarize M1 and M2 macrophages, although it has been argued that this polarization is over simplified and often leads to confusions in both mouse and human (3). One reason why we use the term 'M2-like' here is because oleate-polarized macrophages show certain M2 markers, for instance Retna1, Arg1, Chil3 and Mrc1 (CD206), but also markers of tumor-associated macrophages including IL-6 and VEGFa and even some classical M1 markers for instance iNOS and TNF α . More importantly, as we published before, those oleate polarized cells are immune suppressive. In another study, Hossain and colleagues also found an elevated fatty acid oxidation in suppressive myeloid cells (MDSC) in a tumor model, which was compromised by etomoxir treatment (4). Therefore, we assume that elevated mitochondrial respiration might be a common feature for suppressive myeloid cells, and we prefer to name oleate-polarized myeloid cells rather TAM-like than M2 macrophages. Secondly, as we published before (5), oleate-induced suppression of MSC-2 cell line is independent of IL-4. Also, in the present study, the polarization of bone marrow cells works perfectly in the presence of IL-4 blocking antibodies (Figure 1). We suggest that there are at least two distinct signalling pathways to polarize anti-inflammatory macrophages e.g. IL-4-STAT6 pathway and oleate-mTORC2 pathway.

2) The side effect of etomoxir (6-8): Although etomoxir is only one of the five inhibitors we used in our manuscript, we agree that it is important to clarify the effect of etomoxir in our system. As described in the work by Divakaruni (6), etomoxir impacts the homeostasis of CoA, as proven by the rescue of 200 μ M etomoxir's effect on IL-4 polarized M2 macrophages via addition of CoA. Work by Brenda Raud (7) suggests that 3 μ M etomoxir can specifically suppress CPT1a, however above 100 μ M they observed side effects. We used 40 μ M of etomoxir in our experiments, which is five-fold lower than the concentration applied in these publications and also below the side effect threshold of 100 μ M. However, we agree that we cannot entirely exclude said side effects. The side effect of etomoxir includes at least two parts: first the impaired homeostasis of coenzyme A via direct binding (6) and second, the impaired mitochondrial respiration via suppressing the mitochondrial respiratory complex I (8). CoA is the important substrate to synthesis acetyl-CoA, which is used to generate ATP in mitochondrial. Interruption of mitochondrial respiratory complex I will directly impair the mitochondrial respiration as well as ATP production. In our study, 40 μ M etomoxir led to the reduction of mitochondrial respiration, which impeded oleate-induced immune suppression in myeloid cells. A significant reduction of ATP has also been found by all the other inhibitors applied in our manuscript (manuscript Figure 3B). This is in line with our hypothesis: reduced mitochondrial respiration leads to impaired suppressive function in myeloid cells. **For this revision, we tested the effect of etomoxir in different doses in our system.** With these experiments we can demonstrate that the inhibitory effect of etomoxir in oleate-treated CD206+ myeloid cell polarization is indeed dose-dependent (Figure 2). We cannot conclude or exclude a side-effect here. If the insufficient function of 3 μ M etomoxir in our system is related to the missing side effect, this result indicates that Cpt1a-independent fatty acid oxidation might be essential for mitochondrial respiration as well as the subsequent immune suppressive phenotype in our system. Eukaryotic cells can use peroxisomes for fatty acid oxidation (9). ABCD2 is one of the essential transporters for the import of fatty acids into the peroxisome (10). Our microarray data indicate that the expression of ABCD2 is significantly increased in oleate-treated myeloid cells when compared to controls (GEO database, GSE118080 and now included in the revised version of the manuscript Figure 1B (Lipid metabolism)). All these data support the hypothesis that oleate-induced mitochondrial respiration is important for myeloid cells to polarize and to fulfil their suppressive function, which might rely on peroxisome-derived fatty acid oxidation but not Cpt1a-mediated fatty acid entry. **Thus, it is possible that Cpt1a-mediated fatty acid transport is irrelevant to oleate-induced myeloid cell polarization. However, it is incorrect to state that fatty acid oxidation is not essential for oleate treated myeloid cell polarization.** For instance in the work from Erika Pearce published in 2014 (11), they provide data indicating, that lipase in lysosomes controls the polarization of M2 macrophage via fatty acid oxidation. **These data support the concept that cells might have alternative pathways to oxidize fatty acids and to support mitochondrial respiration.** This is a question, which we will certainly try to answer in the future, but which is not within the scope of the manuscript at hand.

Furthermore, the idea that lipid droplets are required for fatty acid oxidation has recently been shown to be incorrect as well.

While we appreciate the reviewer's concern, we have to object to the reviewer's conclusion: We do not state that lipid droplets are required for fatty acid oxidation as such, but that lipid-droplet-dependent fatty acid oxidation is required for the polarizing effect of FFA. Nevertheless, we discuss these references now in the revised version of our manuscript (page 13, line 17-19).

• Cold-Induced Thermogenesis Depends on ATGL-Mediated Lipolysis in Cardiac Muscle, but Not Brown Adipose Tissue.

Schreiber R, Diwoky C, Schoiswohl G, Feiler U, Wongsiriroj N, Abdellatif M, Kolb D, Hoeks J, Kershaw EE, Sedej S, Schrauwen P, Haemmerle G, Zechner R.
Cell Metab. 2017 Nov 7;26(5):753-763.PMID:28988821

• Lipolysis in Brown Adipocytes Is Not Essential for Cold-Induced Thermogenesis in Mice.

Shin H, Ma Y, Chanturiya T, Cao Q, Wang Y, Kadegowda AKG, Jackson R, Rumore D, Xue B, Shi H, Gavrilova O, Yu L.
Cell Metab. 2017 Nov 7;26(5):764-777.PMID: 28988822

Both publications describe experiments regarding thermogenesis in brown adipose fat tissue. Our study however, is neither about thermogenesis nor brown adipose tissue nor adipose tissue at all. Nevertheless, Schreiber et al. state, "other cell types as endothelial or immune cells also express low levels of ATGL", which holds true for the myeloid cells in our study. Furthermore, it is stated, that ATGL-mediated lipolysis is essential for thermogenesis in white adipose tissue during fasting and in heart for full cardiac function. As our data demonstrate, it is also essential for the polarization of suppressive macrophages.

Thermogenesis is driven by proton leak in the mitochondrion mediated by UCP1 and other molecules. In our manuscript, we are discussing the role of lipid droplets on fatty acid oxidation and mitochondrial respiration, as measured with the seahorse analyser. Therefore, these are two different events although both are linked to the mitochondrion. Furthermore, even in brown adipose tissue-KO mice (the germline Shin H et al. used in their paper), there is a strong compensation of glucose and fatty acid uptake when lipid droplet formation is disrupted (Figure 4 in the reference), indicating that lipid droplets play an essential role in combustion during cold exposure.

These publications are neither disproving nor confirming our results; they simply treat a different topic. If anything, they support us by stating that the necessity of ATGL-dependent lipolysis exists for certain cell types or different cell functions, respectively.

We thank the reviewer again for pointing out the unspecific effects of etomoxir and the potential pitfall for our conclusion. We carefully discussed the respective side effects and why they are not critical for our hypothesis in this revision letter as well as in the revised version of our manuscript.

In summary, we conclude that oleate-polarized TAMs are essential for immunosuppression in tumor conditions, which is mediated by lipid droplet-derived fatty acid oxidation and mitochondrial respiration. Several alternative pathways to Cpt1a-dependent long chain fatty acid import exist and might be essential for fatty acid oxidation in our system as for example peroxisomal degradation. However, our data strongly suggest that fatty acid-induced polarization is distinct from IL-4-induced M2 macrophage differentiation. Furthermore, our main points remain unchallenged: disruption of lipid droplet formation via DGAT inhibition inhibits the polarization to suppressive cells *in vitro* and *in vivo* and demonstrates therapeutic potential when delivered specifically to myeloid cells in a tumor model.

References:

1. Camus, G., E. Herker, A. A. Modi, J. T. Haas, H. R. Ramage, R. V. Farese, and M. J. J. o. B. C. Ott. 2013. Diacylglycerol acyltransferase-1 localizes hepatitis C virus NS5A protein to lipid droplets and enhances NS5A interaction with the viral capsid core. 288: 9915-9923.
2. Nguyen, T. B., S. M. Louie, J. R. Daniele, Q. Tran, A. Dillin, R. Zoncu, D. K. Nomura, and J. A. J. D. c. Olzmann. 2017. DGAT1-dependent lipid droplet biogenesis protects mitochondrial function during starvation-induced autophagy. 42: 9-21. e25.
3. Martinez, F. O., and S. Gordon. 2014. The M1 and M2 paradigm of macrophage activation: time for reassessment. *F1000prime reports* 6.

4. Hossain, F., A. A. Al-Khami, D. Wyczechowska, C. Hernandez, L. Zheng, K. Reiss, L. Del Valle, J. Trillo-Tinoco, T. Maj, and W. J. C. i. r. Zou. 2015. Inhibition of fatty acid oxidation modulates immunosuppressive functions of myeloid-derived suppressor cells and enhances cancer therapies. *3*: 1236-1247.
5. Wu, H., C. Weidinger, F. Schmidt, J. Keye, M. Friedrich, C. Yerinde, G. Willimsky, Z. Qin, B. Siegmund, and R. J. S. r. Glauen. 2017. Oleate but not stearate induces the regulatory phenotype of myeloid suppressor cells. *7*: 7498.
6. Divakaruni, A. S., W. Y. Hsieh, L. Minarrieta, T. N. Duong, K. K. Kim, B. R. Desousa, A. Y. Andreyev, C. E. Bowman, K. Caradonna, and B. P. J. C. m. Dranka. 2018. Etomoxir inhibits macrophage polarization by disrupting CoA homeostasis. *28*: 490-503. e497.
7. Raud, B., D. G. Roy, A. S. Divakaruni, T. N. Tarasenko, R. Franke, E. H. Ma, B. Samborska, W. Y. Hsieh, A. H. Wong, and P. J. C. m. Stüve. 2018. Etomoxir actions on regulatory and memory T cells are independent of Cpt1a-mediated fatty acid oxidation. *28*: 504-515. e507.
8. Yao, C.-H., G.-Y. Liu, R. Wang, S. H. Moon, R. W. Gross, and G. J. J. P. b. Patti. 2018. Identifying off-target effects of etomoxir reveals that carnitine palmitoyltransferase I is essential for cancer cell proliferation independent of β -oxidation. *16*: e2003782.
9. Hunt, M. C., M. I. Siponen, and S. E. J. B. e. B. A.-M. B. o. D. Alexson. 2012. The emerging role of acyl-CoA thioesterases and acyltransferases in regulating peroxisomal lipid metabolism. *1822*: 1397-1410.
10. Fourcade, S., M. Ruiz, C. Camps, A. Schluter, S. M. Houten, P. A. Mooyer, T. Pàmols, G. Dacremont, R. J. Wanders, M. J. A. J. o. P.-E. Giròs, and Metabolism. 2009. A key role for the peroxisomal ABCD2 transporter in fatty acid homeostasis. *296*: E211-E221.
11. Huang, S. C.-C., B. Everts, Y. Ivanova, D. O'sullivan, M. Nascimento, A. M. Smith, W. Beatty, L. Love-Gregory, W. Y. Lam, and C. M. J. N. i. O'Neill. 2014. Cell-intrinsic lysosomal lipolysis is essential for alternative activation of macrophages. *15*: 846.
12. Kleinfeld, A. M., and C. Okada. 2005. Free fatty acid release from human breast cancer tissue inhibits cytotoxic T-lymphocyte-mediated killing. *Journal of lipid research* *46*: 1983-1990.
13. Petrek, J. A., L. C. Hudgins, M. Ho, D. R. Bajorunas, and J. Hirsch. 1997. Fatty acid composition of adipose tissue, an indication of dietary fatty acids, and breast cancer prognosis. *Journal of clinical oncology : official journal of the American Society of Clinical Oncology* *15*: 1377-1384.

Referee #2 (Remarks for Author):

We like to thank the reviewer for carefully reading the manuscript and for the suggestions for further improvement.

1. Did you see the dose effect of oleate and stearate on the polarization of bone marrow-derived myeloid cells?

Yes, we tested the dose dependent suppression of oleate in the MSC-2 cell line in our previous publication (5). We found 0.8 mM oleate to exert a much stronger effect on MSC-2 cells than 0.2 mM. However, to avoid potential side effect of high dose fatty acids, we used 0.2 mM oleate in this study. Other studies provided evidence that in the tumor tissue the dose of oleate strongly correlates with the progression of the tumor, but is higher than 0.2 mM (12, 13). **As suggested by Referee 2, we tested the polarization of myeloid cells in different doses of oleate and present these data to the reviewer's attention (Figure 3).** Our data demonstrate that oleate induced CD206+ myeloid cell differentiation is dose dependent and confirm that 0.2 mM represents the optimal working concentration. Stearate however, shows no polarizing effect in low doses and becomes toxic in high doses.

2. The results showed that oleate-exposed macrophages suppressed T cell proliferation, and inhibition of lipid droplets pathway in macrophages antagonized such inhibitory effects. Did they also affect the functional activity or markers on T cells?

We thank the reviewer for this important question. **We quantified the tumor infiltrating T cells *in vivo* after control/iDGAT treatment and added these data to the revised version of our manuscript (Figure 4, Manuscript Figure 5F).** Our data suggest that DGAT inhibitor treatment

affects myeloid cells to hamper the infiltration of CD8 T cells into the tumor, ultimately facilitating the anti-tumor immune response as hypothesised.

3. In the discussion, the authors claimed that "analysis of colon cancer patients confirmed the correlation between the accumulation of LDs in TAMs and the clinical stage of tumor." However, these data are not found in the manuscript.

Thank you for the comment. We corrected it as: "Finally, analysis of colon cancer patients confirmed the accumulation of LDs in TAMs."

4. Same subtitles of the first and the second part of the results (page 5 and page 6)?

We corrected this mistake.

5. There are numerous typos, e.g., p14, line 22, "provides a novel anti-tumor strategies", and the manuscript should be carefully checked through.

We corrected the mentioned typos and did carefully proofread the manuscript again, thank you.

Referee #3 (Remarks for Author):

We like to thank the reviewer for reviewing our study and for carefully reading the manuscript.

-pg 7: Replace: "In this context, carnitine palmitoyltransferase 1 (CPT1) controls the import of long chain free fatty acids into the mitochondria via converting coenzyme A into l-carnitine" by "CPT1a catalyzes the transfer of the acyl group of long-chain fatty acid-CoA conjugates onto carnitine, which is an essential step for the mitochondrial uptake of long-chain fatty acids for subsequent beta-oxidation in the mitochondrion".

We performed the requested rephrasing, thank you.

[Unpublished figures for the referees has been removed at the authors' request]

2nd Editorial Decision

30 August 2019

Thank you for the submission of your revised manuscript to EMBO Molecular Medicine, and please accept my apologies for the delay in getting back to you, which is due to the fact that I sought external advice from an expert in the field in order to reach a fair and balanced decision.

Indeed, as you will see from the reports below, your revised manuscript was sent back to referees #1 and #2. While referee #2 is now supportive of publication, referee #1 remains unconvinced that the data adequately support the conclusion. This reviewer regrets the use of chemical inhibitors only and the lack of orthogonal experiments to confirm the results.

As mentioned above, and given these contradictory reports, I contacted an external expert for advice. This adviser stated:

"In my opinion, the manuscript should be published. However, the abstract should be adapted, because the authors do not see that in vivo FAO inhibition reduces tumor growth, while DGAT inhibition does. They should also point out in the abstract the use of inhibitors rather than genetics. If the authors do not wish to tone down the lipid droplet FAO link in the abstract, in vitro and in vivo rescue experiments with acetate should be provided."

Given these considerations, we would like you to discuss the concerns from referee 1 and tone down the text of your manuscript accordingly. If you do have data at hand (rescue experiments), we would be happy for you to include it, however we will not ask you to provide any additional experiments at this stage.

Please provide a letter INCLUDING my comments and the reviewer's reports and your detailed responses to their comments (as Word file).

I look forward to reading a new revised version of your manuscript as soon as possible.

***** Reviewer's comments *****

Referee #1 (Remarks for Author):

The major theme of immunometabolism over the last decade has been that metabolic pathways are not only correlated with immune phenotypes but are in fact instructive towards them. This has been most actively promoted for macrophage M2 polarization. M2 macrophages clearly increase oxidative metabolism while M1 macrophages actively suppress oxidative metabolism. This has been taken one step further by stating that fatty acid oxidation is required for M2 polarization. All of the evidence for this was derived from 1 promiscuous epoxide inhibitor, etomoxir, that is often described as a specific Cpt1 inhibitor. It is not. The field has been dominated by this hypothesis with many high profile papers, reviews, etc. Upon further and more stringent analysis this hypothesis has been shown to be incorrect. Here, Wu et al. state "Here we found that fatty acids, especially unsaturated fatty acids, polarize bone marrow-derived myeloid cells into an M2-like phenotype with a robust suppressive capacity." This is essentially the same hypothesis and evidence used by others. That is, they use only chemical inhibitors at high concentrations and argue that they are specific because others have said so. There is no test for specificity throughout the paper. The problem with this manuscript is not the use of inhibitors per se. Small molecule inhibitors are very important for basic and applied research. The problem is that there are no orthogonal experiments to confirm the results. The paper is inhibitor 1-conclusion, inhibitor 2-conclusion....inhibitor-5 conclusion. The inhibitors do not affect the same pathways and the conclusions are not independently supported by the different inhibitors.

The authors rebuttal does not adequately address these issues. They merely suggest that their macrophages and differentiation is different so experiments in other macrophages or cell types are irrelevant. I find this disingenuous. Clearly fatty acid metabolism in macrophages has an important function. The problem is the authors have not provided stringent experiments to support their conclusions and many known pitfalls have not been addressed.

Minor comments:

The authors state:

"etomoxir was applied to block carnitine palmitoyltransferase 1 (CPT1), which controls the import of long chain free fatty acids into the mitochondrion via converting coenzyme A into l-carnitine." This statement is incorrect on several levels. 1) Cpt1 does not import free fatty acids. They are acyl-CoAs. 2) Cpt1 does not convert CoA into carnitine. THE ENZYME DOES NOT WORK AS THE AUTHORS HAVE DESCRIBED.

The authors state:

"adipose triglyceride lipase (ATGL), hormone-sensitive lipase (HSL) and monoacylglycerol lipase (MAGL) control the export of LDs into the cytoplasm." This statement is incorrect. These enzymes do not control the export of LDs. They are lipases that generate free fatty acids from triglyceride. THE ENZYMES DO NOT WORK AS THE AUTHORS HAVE DESCRIBED.

Referee #2 (Comments on Novelty/Model System for Author):

This study investigated the role of lipid metabolism in regulating macrophage polarization through in vitro and in vivo experimental models. Most of the experiments are well-designed and conclusions were justified. These findings demonstrated the novel role of metabolic substrates in regulating the phenotypes and functions of tissue macrophages, and thus provide new insight into the field. Overall, this is an interesting study with potential translational value.

Referee #2 (Remarks for Author):

The authors have addressed most of the concerns raised by the Reviewers and have improved the manuscript accordingly.

2nd Revision - authors' response

9 September 2019

***** Reviewer's comments *****

Referee #1

We honestly thank the reviewer for thoroughly reading our manuscript and for the critical view on our results.

The major theme of immunometabolism over the last decade has been that metabolic pathways are not only correlated with immune phenotypes but are in fact instructive towards them. This has been most actively promoted for macrophage M2 polarization. M2 macrophages clearly increase oxidative metabolism while M1 macrophages actively suppress oxidative metabolism. This has been taken one step further by stating that fatty acid oxidation is required for M2 polarization. All of the evidence for this was derived from 1 promiscuous epoxide inhibitor, etomoxir, that is often described as a specific Cpt1 inhibitor. It is not. The field has been dominated by this hypothesis with many high profile papers, reviews, etc. Upon further and more stringent analysis this hypothesis has been shown to be incorrect.

Here, Wu et al. state "Here we found that fatty acids, especially unsaturated fatty acids, polarize bone marrow-derived myeloid cells into an M2-like phenotype with a robust suppressive capacity." This is essentially the same hypothesis and evidence used by others.

We politely disagree on that point: we demonstrate the effect of fatty acids on the immunological phenotype of myeloid cells, the dependency on lipid droplets of this process and the therapeutic intervention by targeting specifically tumor-associated macrophages. We do not look into the M1/M2 dichotomy but follow the polarization of myeloid precursors to regulatory macrophages in the presence of unsaturated fatty acids. The metabolic environment shapes directly the immunological phenotype in contrast to an immunological signaling, which then shapes the metabolic state. CPT1-mediated fatty acid oxidation was not the focus of this manuscript.

That is, they use only chemical inhibitors at high concentrations and argue that they are specific because others have said so. There is no test for specificity throughout the paper. The problem with this manuscript is not the use of inhibitors per se. Small molecule inhibitors are very important for basic and applied research. The problem is that there are no orthogonal experiments to confirm the results. The paper is inhibitor 1-conclusion, inhibitor 2-conclusion....inhibitor-5 conclusion. The inhibitors do not affect the same pathways and the conclusions are not independently supported by the different inhibitors.

Yes, we use known inhibitors and yes, etomoxir has meanwhile been proven unspecific. Nevertheless, it seems rather rare, that the off target effect shows such a similarity to the intended effect. Realistically, we do not expect that for all five used chemical inhibitors. And we would like to emphasize again that neither the DGAT1 inhibitor nor the DGAT2 inhibitor do work alone. Only in combination, these inhibitors work as described, strongly suggesting no unspecific effects at work.

One can always confirm certain effects using different methods or different approaches. In this case, where well known chemical inhibitors exist and where all of them, within the lipid droplet-biology, show the same effects when it comes to the immunological phenotype of our cells, this approach still seems sufficient. KO-Mice would also have been an option, but on the one hand, as metabolic effects are quite fluid and we were aiming for a defined time point to switch of the respective enzymes, we would have needed tissue specific, inducible KO-strains (also to avoid compensatory effects), which do not exist for all the enzymes. The usual KO-inducing agents, tamoxifen or poly(I:C), are not very well suited for the work with macrophages due to their immunological effects. On the other hand, using five different KO-mouse strains including crossings in between the

strains would even have caused even legal problems, as “confirmation” is not accepted as justification for animal experiments in Germany. Furthermore, the usage of CRISPR-KO systems would, in our opinion, not have helped with regard to specificities, as off-target effects are also a common problem with that technique. Additionally, transfection of primary macrophages alone adds its own effects to this very sensitive and plastic cell type. Last, as we were aiming for a pharmaceutical intervention and as there are no KO-humans, the chemical inhibitors were an absolute necessity in our project.

Moreover, the manuscript is not about a sequence of inhibitor experiments. The main message is about lipid droplet bearing TAM in the tumor and the tumor microenvironment, about how myeloid cells can be polarized by certain fatty acids alone (without immunological signaling) to mimic functionally the in vivo analyzed cells and how the origin and presence of this cells can be prevented by targeting the lipid droplet formation. Chemical inhibitors were used here to connect the lipid droplet formation to the fatty acid oxidation. We are still convinced that these are important data for our fellow colleagues working in this area and that they are able to interpret these data and these effects based on the methods we used to generate them.

The authors' rebuttal does not adequately address these issues. They merely suggest that their macrophages and differentiation is different so experiments in other macrophages or cell types are irrelevant. I find this disingenuous. Clearly fatty acid metabolism in macrophages has an important function. The problem is the authors have not provided stringent experiments to support their conclusions and many known pitfalls have not been addressed.

While we of course recognize that there are still open questions in our story, we are convinced that with this manuscript we add a novel set of data to the field of tumor and myeloid cell biology. Based on our data, we invite everyone to help to decipher the exact metabolic processes and involved pathways, especially colleagues with more experience in molecular metabolism.

Our hypothesis is clear and all our experiments were planned and performed to guide us towards our conclusion. All results are based on the actual functional phenotype of the cells, not just some generic surface markers, ultimately concluding in an actual change of the actual tumor size. We, as well as our cooperation partners, will of course keep on working on the subject and we are convinced that in the foreseeable future we can deliver new data digging deeper in the metabolic pathways at hand.

The in vitro polarized, so called, M2-cells and the regulatory cells we generate, are indeed very different. We and many others consider the distinct term “M2” as problematic, as it includes many different cell types and it suggests a function based on the expression of a handful of markers, which usually cannot withstand deeper analysis. If macrophages are analysed ex vivo, one can see, that all the markers, which were used about 10 years ago to define “M2 and M1”-cells, are entirely mixed up in different tissues and different physiological or pathogenic states and are just vaguely connected to the actual function (1). In addition, the metabolic states differ and are also not always linked to the dichotomy of pro- or anti-inflammatory cells. That is why we confirmed every experiment in our project directly functionally and that is why it is very difficult to compare cells defined merely on some basic set of markers. We have to emphasize again, that treating macrophage progenitors with M-CSF and the potent cytokine IL-4 or differentiating these cells with GM-CSF and no immunological reactive component but oleate alone, represent very different approaches, resulting in very different cells and most probably differences in the metabolic state as well. We cannot see anything disingenuous in our reasoning.

Minor comments:

The authors state:

"etomoxir was applied to block carnitine palmitoyltransferase 1 (CPT1), which controls the import of long chain free fatty acids into the mitochondrium via converting coenzyme A into l-carnitine." This statement is incorrect on several levels. 1) Cpt1 does not import free fatty acids. They are acyl-CoAs. 2) Cpt1 does not convert CoA into carnitine. **THE ENZYME DOES NOT WORK AS THE AUTHORS HAVE DESCRIBED.**

We thank the reviewer for the correction. We now have written: “etomoxir was applied to block carnitine palmitoyltransferase 1 (CPT1), an enzyme associated with the outer mitochondrial membrane that transfers a long chain acyl group from coenzyme A to carnitine, a process which is required to transport long-chain fatty acids into the mitochondrial matrix(2).”

The authors state:

"adipose triglyceride lipase (ATGL), hormone-sensitive lipase (HSL) and monoacylglycerol lipase (MAGL) control the export of LDs into the cytoplasm." This statement is incorrect. These enzymes do not control the export of LDs. They are lipases that generate free fatty acids from triglyceride. THE ENZYMES DO NOT WORK AS THE AUTHORS HAVE DESCRIBED.

We apologize for the oversimplification. We state now: “adipose triglyceride lipase (ATGL), hormone-sensitive lipase (HSL) and monoacylglycerol lipase (MAGL) facilitate the depletion of lipid droplets upon cell activation. Therefore, ATGL and HSL translocate to the LD membrane and cleave fatty acids from the stored triglycerides and therefore control the degradation of LDs. MAGL converts monoacylglycerols to the free fatty acid and glycerol (3-5)“

Referee #2

Referee #2 (Comments on Novelty/Model System for Author):

This study investigated the role of lipid metabolism in regulating macrophage polarization through in vitro and in vivo experimental models. Most of the experiments are well-designed and conclusions were justified. These findings demonstrated the novel role of metabolic substrates in regulating the phenotypes and functions of tissue macrophages, and thus provide new insight into the field. Overall, this is an interesting study with potential translational value.

Referee #2 (Remarks for Author):

The authors have addressed most of the concerns raised by the Reviewers and have improved the manuscript accordingly.

We thank the reviewer for reading and reviewing our rebuttal and his/her kind words.

References:

1. Mowat, A. M., C. L. Scott, and C. C. Bain. Barrier-tissue macrophages: functional adaptation to environmental challenges. *Nature medicine* 2017. 23: 1258-1270.
2. Yao, C. H., G. Y. Liu, R. Wang, S. H. Moon, R. W. Gross, and G. J. Patti. Identifying off-target effects of etomoxir reveals that carnitine palmitoyltransferase I is essential for cancer cell proliferation independent of beta-oxidation. *PLoS biology* 2018. 16: e2003782.
3. Smirnova, E., E. B. Goldberg, K. S. Makarova, L. Lin, W. J. Brown, and C. L. J. E. r. Jackson. ATGL has a key role in lipid droplet/adiposome degradation in mammalian cells. 2006. 7: 106-113.
4. Wang, H., L. Hu, K. Dalen, H. Dorward, A. Marcinkiewicz, D. Russell, D. Gong, C. Londos, T. Yamaguchi, and C. J. J. o. B. C. Holm. Activation of hormone-sensitive lipase requires two steps, protein phosphorylation and binding to the PAT-1 domain of lipid droplet coat proteins. 2009. 284: 32116-32125.
5. Nomura, D. K., J. Z. Long, S. Niessen, H. S. Hoover, S. W. Ng, and B. F. Cravatt. Monoacylglycerol lipase regulates a fatty acid network that promotes cancer pathogenesis. *Cell* 2010. 140: 49-61.

YOU MUST COMPLETE ALL CELLS WITH A PINK BACKGROUND ↓
PLEASE NOTE THAT THIS CHECKLIST WILL BE PUBLISHED ALONGSIDE YOUR PAPER

Corresponding Author Name: Rainier Glauben, Zhihai Qin

Journal Submitted to: EMBO Mol Med

Manuscript Number: EMM-2019-10698